# Phylogenetic profiling resolves early emergence of PRC2 and illuminates its functional core

Abdoallah Sharaf[1,2] , Mallika Vijayanathan[1], Miroslav Oborník[3,4], Iva Mozgová[1,4]

**Polycomb repressive complex 2 (PRC2) is involved in maintaining transcriptionally silent chromatin states through methylating lysine 27 of histone H3 by the catalytic subunit enhancer of zeste [E(z)]. Here, we report the diversity of PRC2 core subunit proteins in different eukaryotic supergroups with emphasis on the early-diverged lineages and explore the molecular evolution of PRC2 subunits by phylogenetics. For the first time, we identify the putative ortholog of E(z) in Discoba, a lineage hypothetically proximal to the eukaryotic root, strongly supporting emergence of PRC2 before the diversification of eukaryotes. Analyzing 283 species, we robustly detect a common presence of E(z) and ESC, indicating a conserved functional core. Full-length Su(z)12 orthologs were identified in some lineages and species only, indicating, nonexclusively, high divergence of VEFS-Box–containing Su(z)12-like proteins, functional convergence of sequence-unrelated proteins, or Su(z)12 dispensability. Our results trace E(z) evolution within the SET-domain protein family, proposing a substrate specificity shift during E(z) evolution based on SET-domain and H3 histone interaction prediction.**

## Introduction

Polycomb group (PcG) proteins are evolutionarily conserved chromatin-associated multisubunit complexes that are involved in the regulation of key developmental programs in multicellular organisms. PcG proteins were first discovered as regulators of homeotic (*HOX*) gene transcription and developmental body patterning in *Drosophila melanogaster* (Schuettengruber et al, 2017). The best studied PcG complexes are represented by the ubiquitin-ligase polycomb repressive complex 1 (PRC1) and the histone methyltransferase PRC2 that establish histone 2A ubiquitination (H2Aub) and H3 lysine 27 methylation (H3K27me), respectively. PRC1 is hypothesized to have emerged through convergent evolution as its core subunits differ in animals (Gahan et al, 2020) and in plants (Berke & Snel, 2014; Chen et al, 2016) despite

conserved catalytic activity (Calonje, 2014; Schuettengruber et al, 2017). In contrast, PRC2 is hypothesized to have diverged early in eukaryotic evolution, and its core subunits are generally conserved in animals, plants, and other major eukaryotic lineages (Shaver et al, 2010; Baile et al, 2021). By targeting different gene sets in time and space, PRC2 represents a major evolutionarily conserved epigenetic repressive system that governs cell identity and development in multicellular eukaryotes (Mozgova et al, 2015; Piunti & Shilatifard, 2021).

The core of PRC2 comprises four subunits, all of which are conserved in plants and animals (Bauer et al, 2016). Interestingly, both plants and animals make use of multiple alternative versions of some of the subunits, which can confer diverse properties and/or functions to the complexes they form (Margueron & Reinberg, 2011). The fact that PRC2 components are well conserved in multicellular eukaryotes and absent in unicellular yeast models (*Saccharomyces cerevisiae* and *Schizosaccharomyces pombe*) initially indicated that PRC2 may have coevolved with multicellularity (Köhler & Villar, 2008). This notion has since been challenged by the identification of PRC2 subunits and H3K27 methylation in several unicellular eukaryotes spanning different eukaryotic supergroups (Liu et al, 2007; Shaver et al, 2010; Lhuillier-Akakpo et al, 2014; Dumesic et al, 2015; Mikulski et al, 2017; Zhao et al, 2020). In unicellular species, PRC2 has been functionally connected to repression of transposable elements (Shaver et al, 2010; Frapporti et al, 2019), to repetitive DNA elimination in ciliates (Liu et al, 2007; Lhuillier-Akakpo et al, 2014), and to determination of cell identity (Zhao et al, 2021).

In *D. melanogaster*, PRC2 core is composed of four protein subunits: the catalytic subunit Enhancer of zeste [E(z)]—a histone methyltransferase (HMT) responsible for H3K27 methylation, the suppressor of zeste 12 [Su(z)12]—C2H2-type zinc finger protein, and two different WD40 repeat (WDR) domain proteins, namely extra sex combs (ESC) and nucleosome remodeling factor (NURF55) (Müller & Verrijzer, 2009). NURF55 is involved in multiple chromatin-related protein complexes (Hennig et al, 2005; Suganuma et al, 2008). Thus, only E(z), ESC, and Su(z)12 subunits were used to infer the phylogeny of the PRC2 complex to bypass confusing interpretations

[1]Biology Centre, Czech Academy of Sciences, Institute of Plant Molecular Biology, České Budějovice, Czech Republic    [2]Genetic Department, Faculty of Agriculture, Ain Shams University, Cairo, Egypt    [3]Biology Centre, Czech Academy of Sciences, Institute of Parasitology, České Budějovice, Czech Republic    [4]University of South Bohemia, Faculty of Science, České Budějovice, Czech Republic

Correspondence: iva.mozgova@umbr.cas.cz; abdoallah.sharaf@umbr.cas.cz

(Shaver et al, 2010; Huang et al, 2017). In metazoans, PRC2 also acts as a H3K27 mono- and di-methyltransferase, and H3K27 tri-methyltransferase activity of the PRC2 can be enhanced by the presence of extra subunits such as polycomb-like (Pcl) (Müller & Verrijzer, 2009). In plants, however, PRC2 mediates H3K27 tri-methylation, whereas mono-methylation (at least in [peri] centromeric regions) is carried out by the ARABIDOPSIS TRITHORAX-RELATED proteins ATXR5 and ATXR6 (Jacob et al, 2009), acting separately from PRC2. H3K27me3, the conserved hallmark of PRC2 activity, is generally not only connected to transcriptional repression of genes located within facultative heterochromatin (Wiles & Selker, 2017; Chammas et al, 2020) but also repetitive sequences within constitutive heterochromatin in some species (Déléris et al, 2021; Vijayanathan et al, 2022).

The catalytic activity of PRC2 is carried out by E(z), which is therefore the defining functional subunit. E(z) catalytic activity is conferred by its structurally conserved SET (Su(var)3-9, Enhancer of zeste, and Trithorax) domain of 120–130 amino acids in length, which was first identified in several chromatin-associated proteins (Dillon et al, 2005; Zhang & Ma, 2012). The SET-domain structure contains β-strands, several α-helices, turns, and numerous loops (Yeates, 2002). SET-domain–containing HMT's have been organized into seven subfamilies characterized by their phylogenetic relationships: E(z), Trx, ASH, SETD, SMYD, PRDM, and SUV (Dillon et al, 2005; Zhang & Ma, 2012; Zhou et al, 2020). The SUV and E(z) subfamilies are involved in gene repression; the Trx, SETD, and Ash subfamilies are positive regulators; and the members of SMYD and PRDM subfamilies mediate both gene repression and activation (Dillon et al, 2005; Zhang & Ma, 2012; Carvalho et al, 2013; Vervoort et al, 2016; Tracy et al, 2018).

SET-domain proteins were also identified in Bacteria and Archaea. Initially, these were described as a paradigm for horizontal gene transfer from eukaryote hosts to symbiotic or pathogenic prokaryotes (Stephens et al, 1998; Aravind & Iyer, 2003). However, the availability of sequenced bacterial genomes uncovered the presence of SET-domain proteins also in free-living species and environmental samples (Alvarez-Venegas et al, 2007; Alvarez-Venegas, 2014), indicating ancient evolutionary origin of SET-domain proteins. Archaeal SET-domain proteins selectively methylate the archaeal histone H4 at lysine 37, suggesting the existence of chromatin methylation before the separation of the archaeal and eukaryotic domains (Alvarez-Venegas, 2014). Yet, the presence of SET-domain proteins in the Asgard (or Asgardarchaeota) group, a separate domain of life representing the closest prokaryotic relatives of eukaryotes (Eme et al, 2017), has not been determined.

Here, using an automated computational pipeline "PcG-finder," we identify the divergent homologs and reconstruct the phylogeny of PRC2 core subunits in all eukaryotic lineages, including several unicellular organisms hypothesized to be proximal to the root of eukaryotic supergroups. We also determine a likely evolutionary emergence point for each of the PRC2 core subunits and identify the prevailing co-occurrence of E(z) and ESC homologs, proposing that these subunits define ancestral conserved functional core of PRC2. For the first time, we investigate the SET-domain protein family evolution within all domains of life (Archaea, Bacteria, and Eukaryota), including the new subgroup Archaea-Asgard, emphasizing the structure–function

relationship of the E(z)-SET domains using protein structure prediction. For the first time, we identify E(z) orthologs in the early diverged group Discoba (Adl et al, 2019), providing significant support for PRC2 emergence before the diversification of eukaryotic lineages.

## Results and Discussion

### Specificity and sensitivity of PcG-finder, an automated protein homolog identification pipeline

To identify orthologs of PRC2 subunits in different eukaryotic supergroups, we developed an automated homolog identification pipeline (PcG-finder). We validated PcG-finder by comparing the orthologs identified by PcG-finder in eight organisms from different lineages with previously identified and experimentally validated PRC2 core components (Table S1). Importantly, all the expected E(z) orthologs were specifically identified in all the organisms. Of the eight organisms, our prediction was identical to the expected results for all conserved PRC2 subunits in four organisms of diverse eukaryotic groups (*Arabidopsis thaliana*, *D. melanogaster*, *Caenorhabditis elegans*, and *Cryptococcus neoformans*). PcG-finder identified all the expected E(z), Su(z)12, ESC, and NURF55 homologs in humans (*Homo sapiens*), but two extra NURF55 homologs were identified, which are annotated as extranuclear WD proteins (Table S1). In the fungus *Neurospora crassa*, PcG-finder identified all except the Su(z)12 homologs. In the ciliates *Paramecium tetraurelia* and *Tetrahymena thermophila*, the Su(z)12 and ESC orthologs were not identified by PcG-finder, but more NURF55 orthologs were found. The two NURF55 homologs in *T. thermophila* are annotated as "histone-binding protein RBBP4 or subunit C of the Caf1 complex protein" and are potential candidates for PRC2-related function. The previously identified Su(z)12 orthologs in *N. crassa* and *T. thermophila* that PcG-finder failed to detect have been described as divergent and/or short proteins (Jiao & Liu, 2016; Xu et al, 2021), which is in line with low sequence homology of the full-length but also VEFS-box domain sequences of the respective Su(z)12 proteins (Fig S1A and B). The recently co-purified *P. tetraurelia* Su(z)12-like functional homolog lacks the C2 RBB4-binding domain and the VEFS-Box domain (Pina et al, 2021 *Preprint*), which is likely to impede its identification by a sequence homology-based pipeline. Similarly, the experimentally identified ESC subunits in ciliates are highly divergent in sequence from ESC subunits in other species (Fig S1C). Based on this validation, we conclude that PcG-finder presents a robust sequence homology-based pipeline for proteins with sufficient sequence similarity. Manual curation and experimental validation of PRC2-related function of the identified proteins are needed, particularly for WD-repeat containing proteins (such as the NURF55 orthologs) that may also take part in other complexes. Subunits of high sequence divergence may not be identified, or a modified strategy must be taken (see further results for VEFS-containing proteins). In cases where divergent (sequence unrelated) proteins fulfill the function of canonical PRC2 subunits, such as is the case for suggested Su(z)12 functional orthologs (Bender et al, 2004; Dumesic et al, 2015), these proteins

are likely to be absent in our search and need to be identified experimentally.

## Phylogenetic distribution of putative PRC2 subunit orthologs

PRC2 subunits have been studied in representative unicellular and multicellular species of diverse eukaryotic supergroups (Lhuillier-Akakpo et al, 2014; Huang et al, 2017; Lewis, 2017; Schuettengruber et al, 2017; Ridenour et al, 2020; Bieluszewski et al, 2021; Vijayanathan et al, 2022). These studies, however, mainly focus on model species, where unicellular models and less well-studied eukaryotic lineages are underrepresented. The diversity of Poly-comb and Trithorax complexes in unicellular organisms was recently investigated in species represented in the Marine Microbial Eukaryote Transcriptome Sequencing Project (MMETSP) database (Keeling et al, 2014; Zhao et al, 2020), where major eukaryotic lineages are characterized but several groups, including Metamonada, Discoba (Eutetramitia), Opisthokonta (Metazoa), Viridiplantae (Embryophyta), Rhodelphidia, Colponemidia (Alveolata), and Eustigmatophyceae or Oomycota (Stramenopila) were missing or not studied. In quest for the origin of PRC2 subunits in eukaryotes, we used publicly available genomic and transcriptomic data from species within all eukaryotic lineages including Discoba and Metamonada, groups hypothetically proximal to the root (Burki et al, 2020). We used the full-length reference protein sequences of PRC2 subunits from *D. melanogaster* as a query and retained only those sequences that contained catalytic or conserved domains with similarity to the respective reference sequence. In total, ge-nomes or transcriptomes of 283 species were explored here (Fig 1 and Table S2). Orthologs of at least one of the PRC2 subunits were identified in 262 (92.6%) of these species, whereas all the subunits were found in 56 (19.8%) species (Fig 1 and Table S3). Of 283 species studied, E(z), Su(z)12, ESC, and NURF55 were identified in 114 (~40.3%), 79 (~27.9%), 125 (~44.2%), and 252 (~89%) species, re-spectively (Fig 1 and Tables S4–S7). Although E(z), ESC, and Su(z)12 are bona fide PRC2 subunits in animals and plants (Jacob et al, 2009; Müller & Verrijzer, 2009; Liu et al, 2011; Wiles & Selker, 2017), NURF55 is involved in multiple chromatin-related protein complexes (Hennig et al, 2005; Suganuma et al, 2008), and its contribution to PRC2 needs to be verified experimentally. Here, we therefore based our conclusions mainly on the orthologs of E(z), Su(z)12, and ESC (Fig 1 and Tables S4–S6), with particular focus on the E(z) subunit. Results on Su(z)12, ESC, and NURF55 orthologs are summarized before focusing on E(z) in more detail.

Using PcG-finder, we identified Su(z)12 in most of the eukaryotic lineages (Fig 1). However, it is not found in representatives of Discoba, Metamonada, Amoebozoa, Glaucophyta, and Haptophyta, representative fungal species and many species of Alveolata or Stramenopila (Fig 1 and Table S5). Su(z)12 is generally absent in Alveolata except for the recently described Colponemidia clade (Tikhonenkov et al, 2020), suggesting divergence of the Su(z)12 at the root of Myzozoa. All the identified Su(z)12 orthologs contain the VEFS-Box domain alone or together with sequentially arranged canonical domains of zinc fingers and other domains (Table S5). Interestingly, Su(z)12 was identified in the binucleate dinoflagellate *Kryptoperidinium foliaceum*, which contains an endosymbiont nucleus of the diatom origin (Figueroa et al, 2009) (Fig 1 and Table

S5). Consistent with earlier reports (Huang et al, 2017; Shaver et al, 2010; Zhao et al, 2020), we did not identify Su(z)12 orthologs in multiple species of core chlorophytes (Viridiplantae) (Fig 1 and Table S5). An unrooted phylogenetic maximum likelihood (ML) tree successfully separated the conserved Su(z)12 orthologs of the different eukaryotic lineages (Fig S3A). As we had not found the VEFS-Box–containing Su(z)12 orthologs in *N. crassa* and *T. ther-mophila* (Table S1), we next searched for VEFS-Box–containing proteins in all the 283 species in this study (Fig 1 and Table S2). VEFS-Box–domain is a conserved domain of Su(z)12 that is required for E(z) interaction and to stimulate PRC2 catalytic activity in *Drosophila* (Ketel et al, 2005; Rai et al, 2013). Using this strategy, we identified VEFS-Box–containing proteins (including Su(z)12) in 154 (54.4%) of the 283 species. Of these, 51 (18%) species contained only Su(z)12 orthologs, and 75 (26.5%) species contained only other VEFS-box proteins, whereas both Su(z)12 orthologs and other VEFS-box proteins were found in only 28 (9.9%) species (Table S8). VEFS-Box proteins were also found in representatives of core chlorophytes except for *Volvox carteri* and in representatives of Discoba, Met-amonada, Haptophyta, fungi, and Alveolata (Fig 1 and Table S8). Among the VEFS-Box proteins, we also identified the Su(z)12-like PRC2 component of *N. crassa* (Jamieson et al, 2013) and *T. ther-mophila* (Xu et al, 2021). We did not identify the recently co-purified Su(z)12-like protein from *P. tetraurelia*, supporting the absence of the VEFS-Box domain in this protein (Pina et al, 2021 *Preprint*). Alignment of the VEFS-Box domains sequences for Su(z)12 iden-tified by PcG-finder and described Su(z)12-like orthologs of *N. crassa* or *T. thermophila* shows limited sequence identity (Fig S1B), demonstrating high divergence of VEFS-Box proteins that serve the function of Su(z)12. Su(z)12-like subunit in *N. crassa* in addition contains domains that are not present in Su(z)12 orthologs, such as the plant homeodomain (PHD) (SM000249). Homology search showed that VEFS-Box proteins with such domain organization are specifically found throughout fungal groups (Dothideomycetes, Leotiomycetes, Pezizomycetes and Sordariomycetes) (Fig S2). Al-together, VEFS-Box proteins are found in all major eukaryotic lineages including Discoba and Metamonada, suggesting their ancient origin. Conserved orthologs of Su(z)12 are found in diverse eukaryotic lineages but seem to be missing in species of several major groups, especially Discoba, Metamonada, Amoebozoa, Glaucophyta, Haptophyta, and Alveolata. Function of Su(z)12 may be fulfilled by diverse VEFS-Box Su(z)12–like proteins as in *N. crassa* (Jamieson et al, 2013) or *T. thermophila* (Xu et al, 2021), demon-strating potentially high sequence divergence of VEFS-Box proteins serving the function of Su(z)12 in PRC2. It must be also noted that absence of Su(z)12 or other VEFS-Box proteins does not rule out the possibility that functional but not sequence homologs may fulfill the function of Su(z)12, such as proposed for PRC2 subunits in *C. elegans* or *C. neoformans* (Bender et al, 2004; Dumesic et al, 2015).

ESC orthologs were found in all the major studied lineages except for Amoebozoa (with only two species representatives), indicating general conservation of the subunit (Fig 1 and Table S6). ESC orthologs seem to be missing from most of the alveolates, being retained only in some ciliates, colponemids and *Digyalum* (Fig 1 and Table S6). The identified ESC orthologs clustered into three clades in a ML-based phylogenetic tree (Fig S3B). The constructed ESC phylogenetic tree unresolved several lineages'

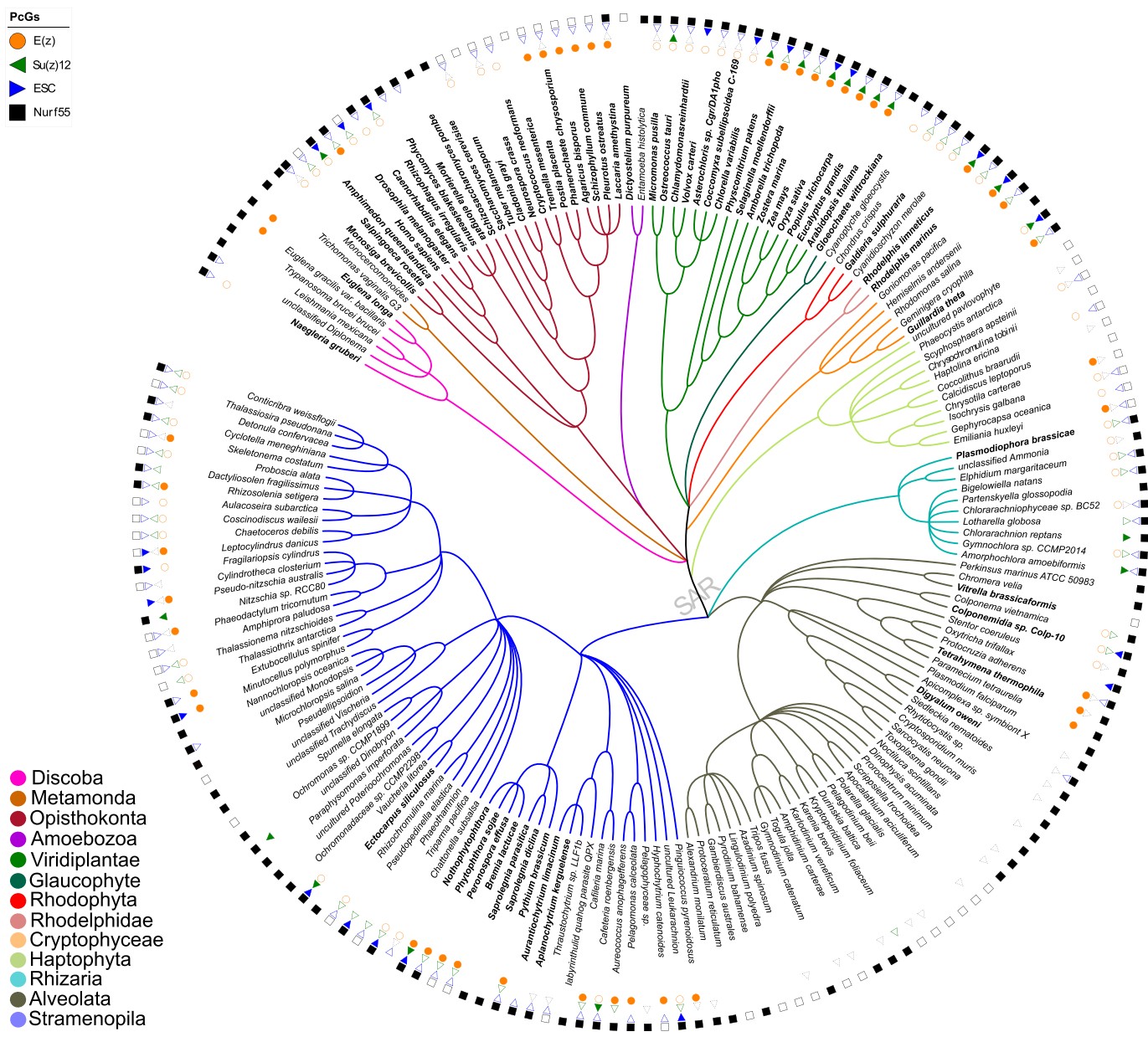

**Figure 1. Species cladogram showing the distribution of PRC2 subunit orthologs identified using PcG-finder across eukaryotic lineages.**
Of the 283 species analyzed using the PcG-finder, only species with the BUSCO score C ≥ 50 (181 species) are plotted (Table S2), and species with score C ≥ 75 (61 species) are marked in bold. Empty symbols indicate one identified homolog, filled symbols indicate multiple identified homologs. Homologs may originate from one or more genomic loci. In case of organisms in which only VEFS-domain (Su(z)12-like) subunits were identified, Su(z)12 is depicted using a gray dotted-line triangle. Cladogram topology is based on the recently proposed topology of the eukaryotic phylogenetic tree (Burki et al, 2020). SAR = Stramenopila, Alveolata, Rhizaria.

relationships, including relationships within microalgae species, as reported previously (Zhao et al, 2020). We identified up to four WD-40 repeat domains in the ESC proteins (Table S6) that can potentially form a platform for protein–protein interactions. As mentioned above, novel or divergent ESC subunits were identified experimentally as PRC2 components in *T. thermophila* and *P. tet-raurelia* (Table S1) (Pina et al, 2021 *Preprint*; Xu et al, 2021). However, these ESC-like proteins display limited sequence similarity to ca-nonical ESC subunits in other species (Fig S1C) and are not re-covered using PcG-finder. Search for divergent ESC using WD40

domain leads to false positive results as WD40 proteins are widely spread. Experimental work will be needed to explore the presence of novel or divergent ESCs.

We detected general presence of NURF55 orthologs that con-tained up to four annotated WD40 domains throughout the eukaryotic lineages (Fig 1 and Table S7). The species distribution in a computed unrooted phylogenetic tree was disordered, indicating complex evolutionary relationships (Fig S3C). Focusing on the known NURF55 orthologs, a separation of *A. thaliana* and *H. sapiens* NURF55 orthologs into two groups was observed. Interestingly, all

the currently known PRC2-involved NURF55 orthologs, including *D. melanogaster* NURF55, human RBBP4 and RBBP7, and *A. thaliana* MSI1, clustered in the better resolved phylogeny crown (Fig S3C and Tables S7 and S1), suggesting common origin of the known PRC2-engaged NURF55 orthologs. Experimental validation will be needed to associate the NURF55 orthologs with PRC2 function and determine whether the phylogenetic position can distinguish orthologs with specific engagement in certain chromatin complexes.

### E(z) subunit is found in all eukaryotic supergroups but is lost in many species

E(z) was identified in representatives of all eukaryotic lineages, except for the studied species of Metamonada and Amoebozoa (Fig 1 and Table S4). In agreement with previous studies, E(z) was identified in all studied members of the green lineage (Viridiplantae) (Shaver et al, 2010; Huang et al, 2017). Interestingly, within the Discoba lineage, E(z) orthologs were identified only in euglenoids and the heterolobosean free-living amoeboflagellate *Naegleria gruberi*. Among alveolates, E(z) orthologs were identified only in the ciliates and in the colponemids (Fig 1 and Table S4). Given the general presence in all eukaryotic supergroups, the absence of E(z) orthologs is likely to occur secondarily. In contrast to previous study by Huang et al (Huang et al, 2017), we identified only one E(z) ortholog in the studied chlorophytes and two or more orthologs in all embryophytes (Fig 1 and Table S4).

Earlier, the absence of PRC2 subunits in some eukaryotic lineages was observed (Zhao et al, 2020) and was attributed to low or undetected expression in the transcriptomic data from the MMETSP database (Keeling et al, 2014). It is not likely to be the case here, at least for some species, because of the high genome sequence coverage in 55% of metamonads and 28% of alveolates studied (Table S2). To further support the potential absence of E(z) in some species compared with others where we found it, we quantitatively evaluated the starting datasets for completeness in terms of expected gene content using BUSCO (https://busco.ezlab.org) (Seppey et al, 2019). The BUSCO scores show that 182 (64.3%) of the 283 used datasets show more than 50% completeness (Fig 1 and Table S2) and are available for representatives of all supergroups. In alveolates, a higher proportion of species with E(z) identified would be expected if E(z) was generally present, strongly suggesting that E(z) is missing from a considerable subset of species within this group. Nevertheless, we cannot rule out the possibility of insufficient data quality or poor structural annotation of these datasets that would impede the identification of homologs in the individual species. As an example, the previously identified E(z) subunit (Shaver et al, 2010; Huang et al, 2017; Mikulski et al, 2017) based on an older *Chlamydomonas reinhardtii* genome assembly (v4) (Merchant et al, 2007), is a short sequence (56 aa) lacking the catalytic SET-domain. However, we could identify a longer E(z) homolog of 793 amino acid in length using the new version of the *C. reinhardtii* genome assembly (v5.6) (O'Donnell et al, 2020). True absence of E(z) in species or lineages will need further experimental confirmation. It will be interesting to identify alternative SET-domain proteins that may take the function of E(z) and/or elucidate alternative pathways that may fulfill the function of H3K27me3-mediated repression. Importantly, we found E(z) orthologs in Discoba, one of the lineages closest to the eukaryotic root

(Burki et al, 2020). Such finding significantly supports the hypothesis of the E(z) presence in the LECA and its emergence preceding the diversification of eukaryotes (Shaver et al, 2010; Zhao et al, 2020). In addition, loss of E(z) seems to be a recurring event in the eukaryotic evolution largely affecting alveolates (Fig 1 and Table S4).

To resolve the evolutionary relationships between E(z) orthologs, we constructed a eukaryotic-E(z) phylogenetic ML tree based on 814 sequences, comprising sequences identified using the PcG-finder and sequences previously identified in KOG1079. An unrooted phylogenetic tree clustered E(z) orthologs into five clades (I–V), where clades I and II were less well distinguished from each other (Fig 2 and Table S4). We did not identify overlap of species between these five clades, except for 19 stramenopile species of clades IV and V E(z) (Oomycota, Ochrophyta, and Hyphochytriomycota) (Fig S4A and Table S9). This finding supports divergent evolution of E(z) subunit orthologs of a single clade within species or even whole eukaryotic lineages, perhaps in connection with the gene duplication during evolution, as was recognized before within green lineage (Huang et al, 2017). The divergence of homologs of the 19 stramenopile species remains unclear. However, it indicates the coexistence of two separate clades of E(z) orthologs in these species. In summary, the phylogenetic tree of eukaryotic E(z) orthologs highlights the existence of five different subclasses of E(z) sequences, which are partially lineage specific.

Having identified five clades of the E(z) orthologs based on the full-length protein sequences, we next asked whether the domain architecture differentiates orthologs in respective clades. From each of the five clades, we selected sequences representing the full diversity of the multiple sequences alignments using hhfilter script from HH-suite3 software (https://github.com/soedinglab/hh-suite) (Fig S5 and Table S10). Domain architecture screening shows that the representative sequences comprise different combinations of the three domains (SET, CXC, and SANT). Clade I and II proteins contain either only the SET domain or sequentially arranged CXC and SET domains (Fig S5 and Table S10). The SANT domain is present in clades III-IV N-terminally of the arranged canonical domains (CXC and SET) (Fig S5 and Table S10). More complex domain architectures were observed within proteins of clade V. This clade contains all plant and animal sequences, including eumetazoan orthologs (EZH1 and EZH2) (Fig S5 and Table S10). Protein domain diversity within E(z) has been observed before within the green lineage (Huang et al, 2017) and other eukaryotic lineages (Shaver et al, 2010; Zhao et al, 2020).

In summary, we identify orthologs of PRC2 core subunits, and particularly E(z) orthologs, in all eukaryotic lineages, including Discoba — a lineage currently placed proximal to the root of eukaryotes (Burki et al, 2020). This indicates the early origin of the PRC2 complex with its possible presence in the LECA. We find a higher number of species where E(z) and ESC orthologs are both identified (95 species) than those where E(z) and canonical Su(z)12 are identified (62 species) (Fig S4B). This suggests that E(z)-ESC likely represents an evolutionarily conserved primary functional core of the PRC2. We have not found conserved Su(z)12 orthologs in multiple lineages including Discoba and Metamonada (Fig 1). VEFS-Box proteins are nevertheless present in most of these lineages (Fig 1 and Table S8), and they may represent more divergent Su(z)12-like proteins. VEFS-Box proteins are found in alveolates where E(z) homologs seem missing (Fig 1). We found non-Su(z)12 VEFS-Box proteins in model species with well-described

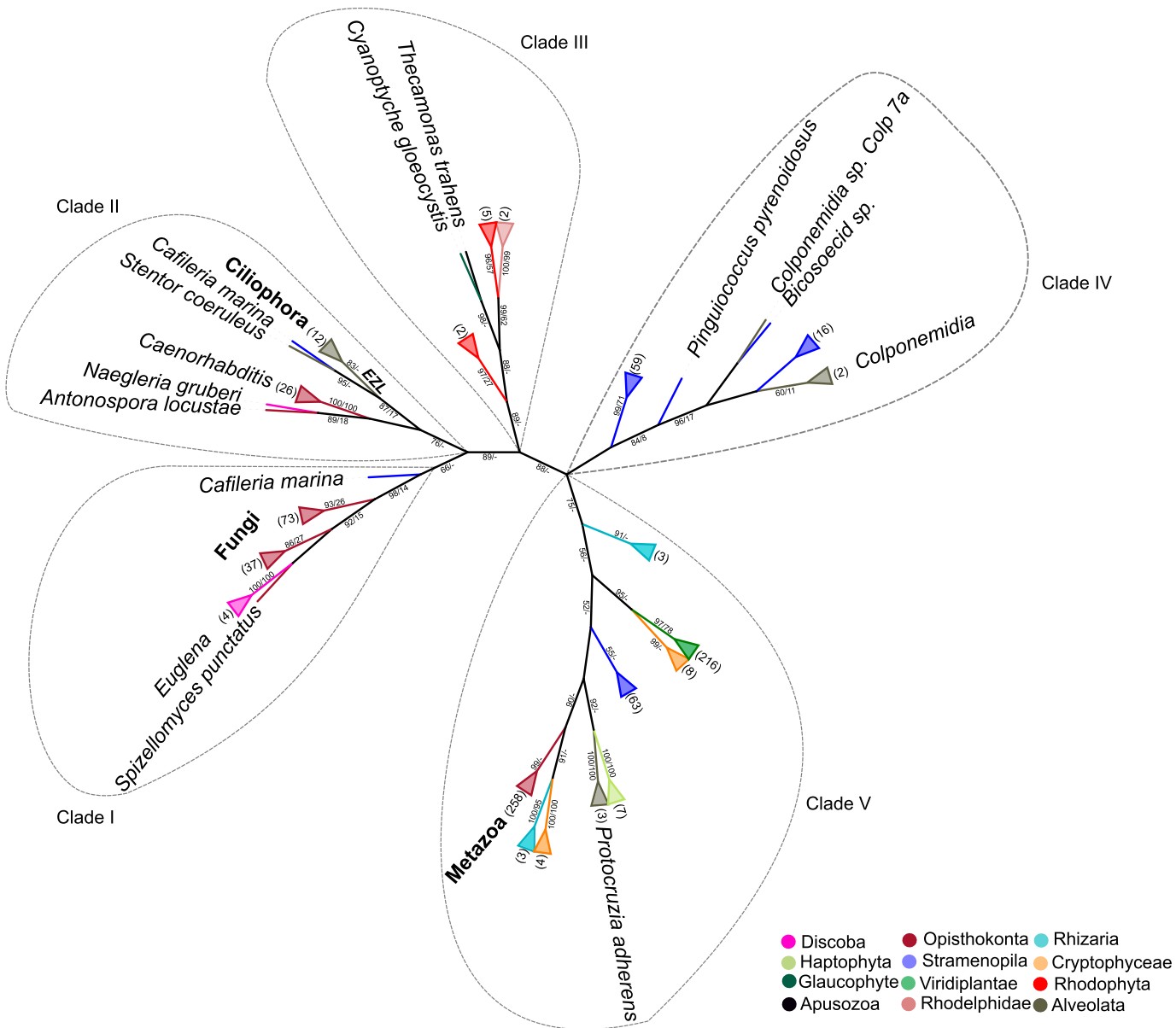

**Figure 2. A maximum likelihood (ML) eukaryotic phylogenetic tree of E(z) orthologs showing separation into five clades (clade I–clade V).**
The tree was constructed using the whole protein sequence and the ML branch support values are given in % (IQ-TREE/RAxML-NG). The number of leaves of the collapsed clade is indicated between parentheses. Higher taxa names are indicated in bold font and species names in normal font.

Su(z)12 orthologs, such as human, *D. melanogaster*, or *A. thaliana*, indicating potential engagement of VEFS-Box proteins in other complexes. Experimental evidence will be needed to associate VEFS-Box proteins with PRC2. It is interesting to note that the currently described examples of PRC2 where other than VEFS-box proteins take the function of Su(z)12 come from species (*C. elegans*, *C. neoformans*, and *P. tetraurelia*) in which no VEFS-Box proteins were identified (Table S8) and which contain ancestral E(z) of clades I or II. Similarly, highly divergent Su(z)12 (or even other VEFS-Box proteins) are involved in PRC2 in fungi and ciliate *T. thermophila* within clade I and/or clade II E(z). This may indicate that other than VEFS-box proteins may be engaged in PRC2 complexes that contain ancestral E(z), but this hypothesis must be

addressed experimentally. Similar to other subunits, the NURF55 orthologs are also well conserved in eukaryotes (Fig 1 and Table S7), and the so-far characterized PRC2-involved NURF55 orthologs seem to form a monophyletic cluster. Involvement of NURF55 in multiple chromatin-modifying complexes impedes concluding on their conserved participation in PRC2, and this aspect will require further study.

**H3K9 methyltransferases branch at the root of E(z) evolution**

To explore the origin of the eukaryotic SET-domain proteins including the E(z), we investigated the SET-domain proteins in

**Figure 3. A maximum likelihood (ML) phylogenetic tree showing the evolutionary relationships of prokaryotic and different eukaryotic subfamilies of SET-domain proteins.**
The tree was constructed using the whole protein sequence, and the ML branch support values are given in % (IQ-TREE/RAxML-NG). The number of leaves of the collapsed clade is mentioned between parentheses. The eukaryotic groups are labeled and the SET-domain subfamilies as well, whereas the uncharacterized SET-domain is not labeled.

prokaryotes and eukaryotes. For this analysis, we used a modified PcG-finder. The hits mapping to the bacterial orthologous group (OG) clustering of the SET-domain protein (COG2940) were selected in the orthologs assignment step. The SET-domain proteins were identified in 41 of 105 bacterial species studied (39%), representing 26 different taxonomic groups (Table S11). We found the SET-domain proteins only in Euryarchaeota and not in other archaeal groups (Table S11). For the first time, we identified a SET-domain protein in one of the Asgard species (*Candidatus Lokiarchaeota archaeon*) (Table S11).

The ML phylogenetic tree was constructed based on the full-length E(z) sequences (Table S4), prokaryotic SET-domain protein sequences (Table S11) identified in this study, and sequences from

different SET-domain protein subfamilies identified previously in *A. thaliana, D. melanogaster*, and *H. sapiens* (Dillon et al, 2005; Zhang & Ma, 2012) (Fig 3). The tree was rooted using the Asgard homolog (TFG06932_1) (Fig 3 and Table S11).

Sequences of SET-domain proteins within each subfamily formed monophyletic groups (Fig 3) as shown previously (Huang et al, 2017). The relationships and topology of the SET-domain subfamilies were highly similar. Interestingly, the E(z) subfamily clustered in the crown of the tree without any connection to the prokaryotic sequences and shows a similar topology as the eukaryotic-E(z) tree (Fig 2). The branch containing E(z) orthologs forms a well-separated sister group to the SET-domain subfamilies SET-SUV, SET-ASH, and SET-Trx. All these subfamilies branch

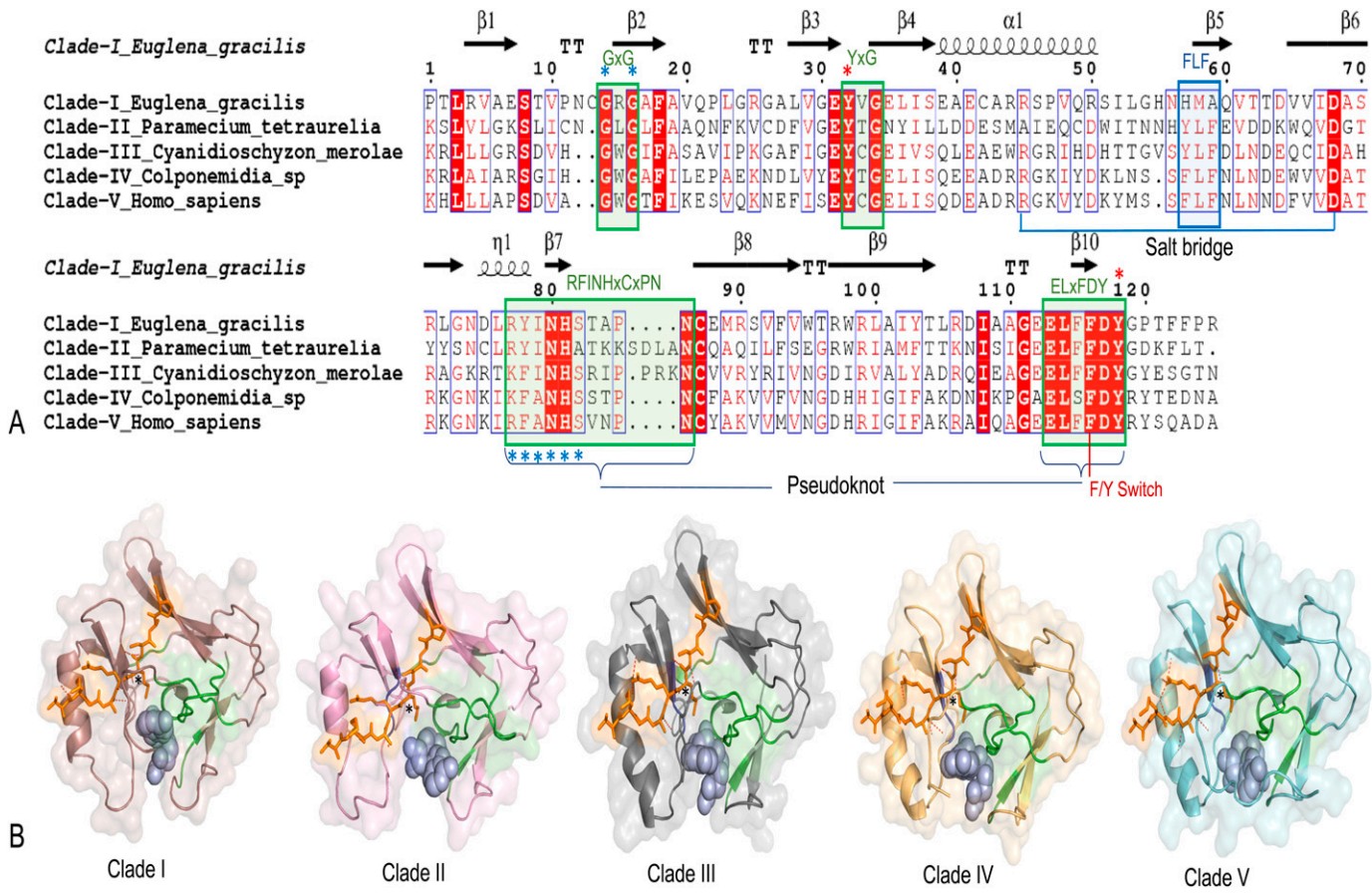

**Figure 4. Sequence structure comparison of representative SET-domains from five E(z) clades.**
**(A)** Alignment of SET-domains of different clade representatives. Invariant (absolutely conserved) residues are highlighted in red background with white text. Conserved residues are highlighted in red font. Conserved signature motifs (GxG, YxG, RFINHxCxPN, and ELxFDY) (Green box) and hydrophobic FLF (contribute to lysine binding pocket) (Blue box), salt bridge that causes intramolecular interaction, pseudoknot, F/Y switch controlling methylation (mono-, di-, or tri-methylation), catalytic sites (red asterisk), cofactor binding sites (blue asterisk) are highlighted. **(B)** Cartoon/surface view of the SET domain of each clade. Human (*H. sapiens*) EZH1 SET-domain (PDB ID: 7KSR, EZH1) was used as a reference model structure. Cofactor (light blue), and the substrate analogs (H3K27M-peptide inhibitor; orange color) are represented by stick and sphere forms. The position of "K27M" and lysine binding channel is indicated by a black asterisk. Peptide-binding cleft area is highlighted in light orange. Conserved signature motifs are highlighted in green and the FLF motif in blue. Interactions of domain with peptides (hydrogen bonds) are represented by red dashes.

separately from the SET-PRDM, SET-SMYD, and SET-SETD subfamilies. The E(z) subfamily of SET-domain proteins therefore seems to have diverged after the emergence of eukaryotes but before their expansion. Within the rooted E(z) phylogeny, we distinguished five clades of E(z) sequences, similar to the unrooted eukaryotic-E(z) tree (Fig 2). The most ancestral clade I includes fungal (Basidiomycetes) and Discoba (*Euglena*) E(z) proteins. Clade II contains ascomycete, the basidiomycete (*C. neoformans*), ciliate, and *C. elegans* E(z) proteins. Rhodophyta and Rhodelphea E(z)s formed clade III; oomycete's, ochrophyte (Stramenopila), and colponemid (Alveolata) sequences constituted clade IV. Clade V was represented by diatom (Stramenopila), Cryptophyceae, Rhizaria, Haptophyceae, plant (Viridiplantae), and metazoan (Ophistokonta) E(z) orthologs (Fig 3 and Table S4).

We addressed the question of whether the tree topology of the SET-domains follows the phylogeny of E(z) orthologs and their separation into five clades. A congruency of topologies may potentially indicate functional separation (e.g., substrate specificity) of the SET-domain proteins. To this end, we extracted the amino

acid sequences of the SET-domains using an in-house Python script (SET extractor). Next, we constructed a rooted-ML tree using the extracted SET-domain sequences (Fig S6) from the sequences used to compute the full-length SET-domain protein phylogeny (Fig 3). Both phylogenetic trees show similar topology; and in particular, the clustering of the E(z) subfamily members into five clades is retained. Still, we identified a few interesting topology differences in the phylogeny of the SET-domains. For instance, the glaucophyte *Cyanoptyche gloeocystis* SET-domain branched in the root of all E(z)-SET-domain sequences. However, because many nodes of the SET-based tree are not supported (Fig S6), the topology can be affected by the low informational content of the relatively short SET-domain. Moreover, the sequence of the alphaproteobacteria *Pelagibacter ubique* clustered with the rhodophyte *Cyanidioschyzon merolae* within clade III that includes all other rhodophytes. This could indicate that in clade III of E(z) orthologs, the SET-domain retained certain bacterial features. Finally, some SMYD-SET domain sequences branched in the root of clade V of E(z) SET-domains (Fig S6).

SET-domains display different substrate specificity (Herz et al, 2013). Interestingly, this also holds true for different SET-domains within the E(z) subfamily as some E(z) homologs have been demonstrated to methylate substrates other than H3K27 (Frapporti et al, 2019). We therefore asked whether the clustering of the SET-domains and the separation of the E(z) SET-domains into the different clades may correlate with substrate specificity. The SET-domain consists of a series of conserved sequence motifs, including GxG, YxG, RFINHxCxPN, and ELxFDY, where the last two motifs form a pseudo-knot–like conformation in the fold that has specific roles in binding and catalysis (Cheng et al, 2005) (Fig 4). This pseudoknot conformation forms an active site close to the cofactor binding pocket. The binding pocket is crucial for the specific interaction of the methyl donor (cofactor) with specific residues of H3 through a narrow hydrophobic deep lysine channel (Dillon et al, 2005).

In general, four sequence motifs (GxG, YxG, RFINHxCxPN, and ELxFDY) are substantially conserved throughout all E(z) SET-domains of the five clades (Figs 4A, S7, and S9A). Despite minor amino acid substitutions, they fundamentally adopt a comparable structural fold with a conserved antiparallel β-barrel and a slightly varied cofactor binding pocket containing the enzyme active site residues, with minor/or no changes in the peptide-binding cleft (Fig 4A and B). We investigated whether the peptide interaction in the catalytic SET-domain could help to distinguish the SET-domains of the five E(z) clades and explain the differences in substrate selectivity. Structural models of proteins belonging to the different E(z) SET-domain clades, either in the apo (no ligand) or holo form (when bound to the cofactor S-adenosyl homocysteine [SAH] and a peptide substrate analog [H3K27M]), were generated to mimic the domain/substrate/cofactor complex (Fig 4B). Except for clade II, the interaction pattern of all E(z)-domain representatives exhibited a similar binding pattern for both cofactor and peptide (Figs 4B and S8). The representative protein model for clade II (*P. tetraurelia*) is comparable to the previously computed model (Frapporti et al, 2019), although the amino acids comprising the C-terminal post-SET domain are not included in our model. Importantly, structural folds are conserved, and retain similar structural topology and substrate binding patterns (Frapporti et al, 2019). Surprisingly, the structure model of the glaucophyte *C. gloeocystis* SET-domain (root of E(z) SET-domains—Figs S6 and S9A) shows better interaction with H3K9M than with H3K27M (Fig S9B). "K9M" was directed toward the channel which suggests a putative possibility of H3K9me catalysis (Fig S9B and D). Previously, H3K9me (and H3K27me) was experimentally shown to be carried out by the Ezl homolog of *P. tetraurelia* (clade II) (Frapporti et al, 2019). Interestingly, the modeled orientation of both the H3K9M and H3K27M in the *P. tetraurelia* SET-domain suggest methylation events (Fig S9C). These findings are in line with the respective phylogenetic positions of these SET-domains at the root of the E(z) SET-domain subfamily in *C. gloeocystis* and early-diverged clade II for *P. tetraurelia* (Figs S3 and S6). It may suggest the possibility of substrate specificity shift from H3K9me to H3K27me during evolution of the E(z) SET-domains.

Despite the intensive investigation of PRC2 core subunits distribution in plants and opisthokonts, the diversity of these subunits within other eukaryotes remains enigmatic. Here, we present phylogenetic profiling of PRC2 core subunits across all eukaryotic lineages using a computational automated tool (PcG-finder), which

can be used in future studies. Altogether, our results strongly suggest that the orthologs of the catalytic subunit E(z) and the core subunits ESC and NURF55 existed in lineages hypothetically closest to the eukaryotic root (Fig 5), in agreement with the standing hypothesis of the presence of PRC2 in the LECA. We identify VEFS-Box proteins in most of the eukaryotic lineages including Discoba, proposing that canonical Su(z)12 evolved later and more rapidly than the other subunits of PRC2. In accordance, we identify the co-occurrence of the orthologs of E(z) and ESC more frequently than E(z) and Su(z)12, proposing that the E(z)-ESC module may have been the initial and evolutionarily conserved PRC2 functional module. NURF55 is the most conserved subunit within all eukaryotic lineages, supporting its involvement in other chromatin-related complexes. Moreover, we could not associate the loss of any of the subunits either with the uni- or multicellularity or lifestyle of species within the studied eukaryotic lineages (Fig 5). However, our findings highlight important future questions that need to be addressed experimentally, including PRC2 catalytic activity, E(z) substrate specificity, involvement of VEFS-Box proteins, and other functional homologs of Su(z)12, and biological function of the PRC2. In addition, future work with more advanced versions of genome assemblies will be needed to settle the question of secondary losses of E(z) in individual species of different lineages or even major eukaryotic lineages such as alveolates (Fig 5).

# Materials and Methods

### Data sources

The complete predicted proteome sequences of studies organisms (Table S1) were obtained from JGI (http://genome.jgi.doe.gov), the Eukaryotic Pathogen database (https://eupathdb.org), and NCBI GenBank (https://ncbi.nlm.nih.gov/). The Marine Microbial Eukaryote Transcriptome Sequencing Project database (MMETSP) (Keeling et al, 2014) and the 1000 Plants (1KP) (Leebens-Mack et al, 2019) were additional sources for predicted proteome sequences inferred from transcriptomic data. Some proteome sequences were retrieved using mining publications (*Rhodelphis marinus*, *Rhodelphis limneticus*, *Euglena gracilis var. Bacillaris*) and metamonads from Leger et al (2017) or kindly made available by our collaborators. In proteome datasets, when two or more protein sequences at the same locus were identical and overlapping, the longest sequence was considered.

### PcG-finder pipeline

The pipeline we established and called "PcG-finder" is composed of three different steps: homology search, orthology assignments, and domains architecture scanning (Fig S10). The whole pipeline was implemented in a single Python program and deposited in GitHub.

The *Drosophila* E(z) (NP_001137932.1) reference amino acid sequence was used to search through all predicted proteome sequences retrieved previously using the hidden Markov model (HMM)–based tool jackhammer (https://github.com/EddyRivasLab/hmmer). The highest scoring protein from each target organism was then used for

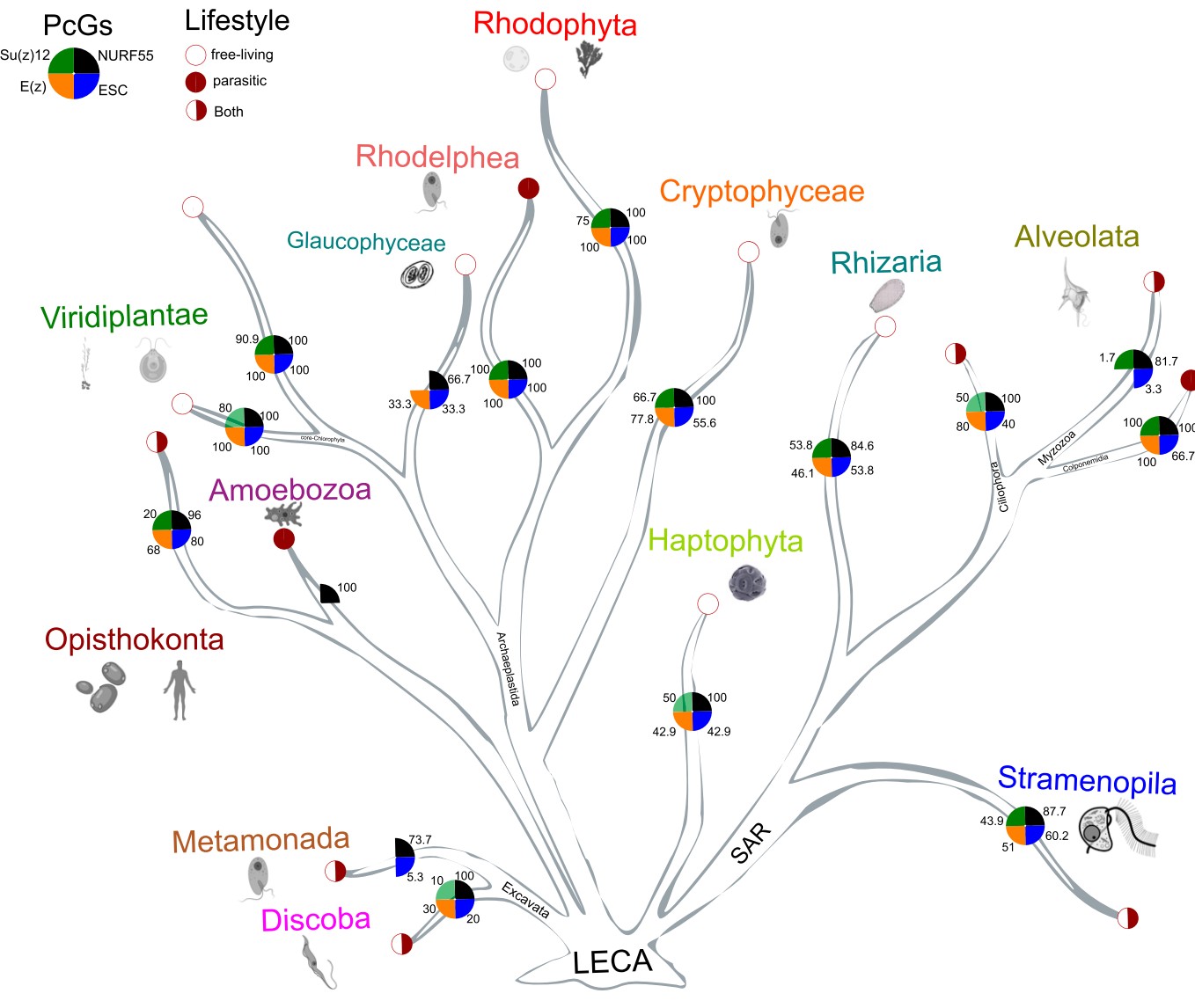

**Figure 5. Schematic of the eukaryotic tree of life shows the summary of PRC2 subunit diversity.**
The conservation of each subunit within a lineage is indicated as percentage of analyzed species in which the subunit was identified. The list (and total number) of species analyzed in each lineage is given in Table S2. In the case of lineages/groups where only VEFS-Box proteins but no Su(z)12 orthologs were identified, the Su(z)12 sector is depicted in transparent green. The depicted tree topology is designed based on the recently proposed topology of the eukaryotic phylogenetic tree (Burki et al, 2020).

a jackhmmer search against the reference genomes of *A. thaliana, D. melanogaster*, and *H. sapiens*, to confirm a reciprocal best match to the original protein. For the other members of PRC2, jackhmmer searches were performed using *Drosophila* reference amino acid sequences Su(z)12 (NP_730465.1), ESC (NP_477431.1), and NURF55 (NP_524354.1) (Fig S10).

Evolutionary genealogy of genes: Non-supervised Orthologous Groups (eggNOG) mapper was used for hierarchical resolution of orthology assignments and fine-grained relationships based on phylogenetic analysis. eggNOG mapper implements a species-aware clustering algorithm based on the concept of triangulation of best reciprocal hits and is applied to identify OGs: sets of homologous sequences that started diverging from the same speciation event. In addition, manually curated OGs are available for the eukaryotes (KOGs) were integrated into the corresponding levels in eggNOG (http://eggnog5.embl.de). Only eggNOG-hits with KOG1079, KOG2350, KOG1034, and KOG0264 for E(z), ESC, Su(z)12, and NURF55, respectively, were selected. These KOGs were identified earlier using *Drosophila*'s reference amino acid sequences (Fig S10).

Finally, the SMART (http://smart.embl.de) and Pfam (http://pfam.xfam.org) databases were employed to identify conserved domains present in E(z), ESC, Su(z)12, and NURF55 from different organisms; both SMART and Pfam databases were merged; and redundant domains were filtered-out, and the hidden Markov model (HMM)–based tool hmmscan (https://github.com/EddyRivasLab/hmmer)

was used to scan domains architecture. Only sequences with the catalytic or conserved domain of the references were retained.

### Phylogenetic analyses

PcG-finder–identified sequences were supplemented with KOG sequences, and all sequences were aligned using MAFFT software (https://mafft.cbrc.jp/alignment/software/) and ambiguously aligned regions were excluded from further analysis using trimAl software (http://trimal.cgenomics.org). Alignments were tested using ProtTest v3 (https://github.com/ddarriba/prottest3) to choose the appropriate model for amino acid substitution.

Two separated maximum likelihood (ML) phylogenetic trees were computed using RAxML-NG (Kozlov et al, 2019) and IQ-TREE 2 (http://www.iqtree.org) software. ML analyses were performed using 1,000 bootstrap replicates. The supporting values from both pieces of software were noted on the ML-unrooted tree. Only SET-protein phylogenetic trees were rooted with the Asgard sequence, which was considered as the out-group.

### Sequence structure analysis

ESpript v. 3.0 (https://espript.ibcp.fr) was used to visualize the sequence conservation pattern. To get insights into the structural features, protein models were generated for representative species by I-TASSER (https://zhanggroup.org/I-TASSER/) and SWISS-MODEL (https://swissmodel.expasy.org/). Human EZH1 SET-domain was used as a template (PDB ID: 7KSR). SwissDock (http://www.swissdock.ch), and the HawkDock server (http://cadd.zju.edu.cn/hawkdock/) was used for docking, and PyMOL v. 2.4.1 (https://pymol.org/2/) was used for molecular visualization and comparative analysis between the modeled and reference protein structures.

## Data Availability

All data analyzed in this study are publicly available. PcG-finder pipeline Python software is available at https://github.com/Iva-Mozgova-Lab/PcG_finder, and SET extractor script is available at https://github.com/Iva-Mozgova-Lab/SET_extractor.

## Supplementary Information

## Acknowledgments

This work was supported by the Czech Academy of Sciences, ERC-CZ, grant number ERC200961901 to I Mozgová, and ERDF/ESF, grant number CZ.02.1.01/0.0/0.0/16_019/0000759 to M Oborník. The authors thank Dr. Aleš Horák for providing the transcriptomic data for the some unidentified diplonema and stramenopile species. Also, the authors thank the CESNET LM2015042 and the CERIT Scientific Cloud LM2015085, funded under the programme "Projects of Large Research, Development, and Innovations Infrastructures" for providing the computational resources. We would also like to thank the anonymous reviewers for comments that have improved the original manuscript.

## Author Contributions

A Sharaf: conceptualization, data curation, validation, investigation, visualization, methodology, and writing—original draft, review, and editing.
M Vijayanathan: investigation, visualization, methodology, and writing—original draft.
M Obornik: conceptualization, validation, and writing—review and editing.
I Mozgova: conceptualization, funding acquisition, validation, investigation, and writing—original draft, review, and editing.

## Conflict of Interest Statement

The authors declare that they have no conflict of interest.

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
