## [Reviewer comments · Life Science Alliance]

Life Science Alliance

Phylogenetic profiling resolves early emergence of PRC2 and illuminates its functional core

Abdoallah Sharaf, Mallika Vijayanathan, Miroslav Obornik, and Iva Mozgova

DOI: <https://doi.org/10.26508/lsa.202101271>

Corresponding author(s): *Abdoallah Sharaf, Biology Centre and Iva Mozgova, Biology Centre*

Review Timeline:

Submission Date:	2021-10-22
Editorial Decision:	2021-12-01
Revision Received:	2022-02-18
Editorial Decision:	2022-03-15
Revision Received:	2022-03-21
Accepted:	2022-03-23

Scientific Editor: Novella Guidi

Transaction Report:

December 1, 2021

Re: Life Science Alliance manuscript #LSA-2021-01271-T

Dr. Abdoallah Sharaf
Biology Centre
Plant Epigenetics
Branišovská 31
České Budějovice, NA 37005
Czech Republic

Dear Dr. Sharaf,

Thank you for submitting your manuscript entitled "Phylogenetic profiling resolves early emergence of PRC 2 and illuminates its functional core" to Life Science Alliance. The manuscript was assessed by expert reviewers, whose comments are appended to this letter. As you will note from the reviewers' comments below, all the reviewers are quite positive and think that this study represents a comprehensive phylogenetic profiling of the PRC2 subunits across the entire Eukaryota. They, however, do raise some concerns that would need to be addressed in the revised version before resubmission. We, thus, encourage you to submit a revised version of the manuscript back to LSA that responds to all of the reviewers' points including validating how robust and sensitive the PcG-finder is by showing that you manage to recover all the known PRC2 core components in the organisms where PRC2 composition has been established experimentally as suggested by Reviewer 1, and simplify the presentation of the data in figures 1,2 and 3 in a way that the biological meaning can be delivered to more broad and general readers, as suggested by Reviewer 2 and 3.

Thank you for this interesting contribution to Life Science Alliance. We are looking forward to receiving your revised manuscript.

Sincerely,

-- Summary blurb (enter in submission system): A short text summarizing in a single sentence the study (max. 200 characters including spaces). This text is used in conjunction with the titles of papers, hence should be informative and complementary to the title and running title. It should describe the context and significance of the findings for a general readership; it should be

written in the present tense and refer to the work in the third person. Author names should not be mentioned.

B. MANUSCRIPT ORGANIZATION AND FORMATTING:

Reviewer #1 (Comments to the Authors (Required)):

The manuscript by Sharaf et al examines the evolutionary origin of the Polycomb Repressive Complex 2 (PRC2) using a phylogenetic approach. PRC2 deposits the histone mark H3K27me3 on silent protein-coding genes. The composition of the complex has been characterized in very few species and its enzymatic activity was experimentally tested in a handful of organisms. Here, the authors survey the genomes/transcriptomes of 283 species to search for 4 PRC2 core components: E(z), the catalytic subunit, ESC, SUZ12, and Nurf55. They identified E(z) and Esc homologs in species that support the emergence of PRC2 prior to the diversification of eukaryotes. Their analysis also suggests that the E(z) enzyme may have shifted substrate specificity- from K9 to K27 methylation- in the course of evolution.

Major comments:

1. The authors developed a sequence homology-based computational automated tool (PcG-finder) for the search of PRC2 core components in genomes and transcriptomes, using *Drosophila melanogaster* PRC2 protein sequences. I wonder how robust and sensitive this tool is. Could the authors validate their tool by showing that they manage to recover all the known PRC2 core components in the organisms where PRC2 composition has been established experimentally: *Arabidopsis*, *C. elegans*, *Neurospora crassa*, *Cryptococcus neoformans*, *Paramecium tetraurelia*, *Tetrahymena thermophila*.
2. Along the same line, SUZ12 is not always found in PRC2 complexes. In the fungus *Cryptococcus neoformans* (Dumesic, 2015) and in *C. elegans* (Bender 2004; Xu 2001), no obvious SUZ12 homologs were identified, yet *Cryptococcus* Bnd and *C. elegans* MES3, which are required for the catalytic activity of the complex, might be functional homologs of SUZ12, as suggested before for MES3 (Schuettengruber, 2017). Do the authors identify those proteins with their tool? Pull down experiments recently revealed the existence of putative SUZ12 components in PRC2 in the ciliates *Paramecium tetraurelia* (Miro Pina 2021 <https://doi.org/10.1101/2021.08.12.456067>) and *Tetrahymena thermophila* (Xu 2021 <https://doi.org/10.1093/nar/gkaa1262>) that could not be identified by homology-based search because of sequence divergence. The conclusion that there is evidence of the absence of SUZ12 in ciliates (Figure 1- Figure 5- line 143) should thus be corrected. In addition, the limitations of the approach presented in the manuscript should be discussed. It might also be wise not to draw any conclusions on the dispensability of SUZ12 based on its absence. Instead, I suggest considering the alternative possibility that SUZ12 might evolve more rapidly than the other components. The authors should rephrase the abstract (Lines 21-22) and the results section (lines 143- lines 279-281).

Minor comments:

- Line 13: "maintaining" not "establishing transcriptionally silent chromatin states"
- Lines 50-53: regarding the evidence for H3K27me3 in unicellular eukaryotes, some references are missing (eg Carlier, 2021; Jamieson 2013, Lhuillier-Akakpo 2014; Liu 2007)
- Lines 68-69: The association between H3K27me3 and genomic repeats could be supported by a more complete list of references and/or a review.
- Lines 90-91: The sentence is incomplete: "suggesting the evidence..."
- Lines 132-134: The statement regarding NURF55 function outside PRC2 is correct. It would be better to state that NURF55 belongs to several complexes earlier in the introduction.
- Line 220: reference to E(z) phylogeny in ciliates should be corrected. It is not Frapporti 2019 but Lhuillier-Akakpo 2014.
- Lines 329-332. The sentence is unclear. Why is this a good model for studying the evolution of PRC2 catalytic function?
- Lines 387-389: Regarding the structural modeling of SET domains of E(z) proteins belonging to the 5 clades, the authors state that clade II is different from the others: "some of the loops/turns near the cofactor binding pocket are slightly disordered when

compared to other models, which might be the reason for the binding difference". Could the authors be more precise: I don't understand from their description what exactly differs in clade II enzyme and how this is linked to substrate specificity? Clade II *Paramecium tetraurelia* SET domain has been modeled in a previous study (Frapporti 2019). How similar/different are the two modeled structures and conclusions?

- Lines 392-394: add the reference.

- Lines 415/416: what do the authors mean by "biological significance of PRC2": its biological functions?

- Figure 5: A "parasitism lifestyle" cannot be used to describe Ciliophora/Alveolata, as indicated in the legend. This should be corrected.

Reviewer #2 (Comments to the Authors (Required)):

Polycomb Repressive Complex 2 (PRC2) has been shown to play crucial roles in cell differentiation and cell fate maintenance during the development of many metazoan models by generating specific histone modifications and establishing a poised gene expression program.

In the article, "Phylogenetic profiling resolves early emergence of PRC2 and illuminates its functional core", the authors performed comprehensive phylogenetic analysis to study the distribution of PRC2 components across eukaryotes. Upon surveying the sequence information of 283 species, the authors identified a putative ortholog of Ez in a subset of Discoba, an earlier eukaryotic lineage. The authors did not observe Ez in Metamonada, although both Discoba and Metamonada belong to the Excavates superfamily. After studying distribution, the authors focused on the Ez component, which harbors the enzymatic SET domain thus compose a catalytic part of the PRC2. The authors classified full-length Ez into five groups and identified a group where Discoba is located with other species (group II). Then, the authors zoomed into SET-domain and further compared domain structures across five groups. In a structure model, the histone substrate-binding pocket of group II showed more disordered features than other groups.

Overall, this study represents comprehensive phylogenetic profiling of the PRC2 subunits across the entire Eukaryota. My main concern is the presentation, in particular Figures 1-3, needs to be simplified and up to the point.

1. Figure 1: Manuscript (written part of Figure 1) is relatively straightforward, but Figure 1 itself is too complicated to understand by looking at it. Figure legend is poorly described and contains little information. What are the main messages of the figure? Where is that info? There are many unnecessary parts of the figure as well. For example, what is the added value of the numbers next to the species? Species names are barely readable on a 100% scale. Do you need all those names and numbers in the main figure? The authors should move the current version into supplementary and provide a simple figure. Also, it would be helpful to add a photo or image of each representative species used in their analysis.
2. Among the PRC2 components, NURF55 shows the most substantial conservation across eukaryotes. Nevertheless. The authors focused on Ez. The rationale for Ez selection has to be apparent from the beginning of the manuscript.
3. Figures 2 and 3 have the same issue as Figure 1. The numbers in triangles and branches are too small to recognize and may not be necessary for the main figure. Figure 2A - Information about different species in each clade is an interesting biological outcome, but it is hard to get from the figure. Figure 2B - it is unclear what is the take-home message of this figure. The written part (page 7; 251-272) is also confusing.
4. I understand the challenge of assembling and organizing all info after sequencing analysis. However, the current version of the figure set seems premature. These figures show the crude output of the research (species names, conservation scores, etc.), and the authors made a little effort to deliver biological meaning to more broad and general readers.
5. Figure 4 is excellently delivered the biological message. Figure 5 is constructive to get the summary view of Figure 1-3. It would be helpful to add info about the percentage value of the existing PRC2 components. For instance, most Alveolata they observed do not contain Suz12, ESC, and Ez.

Reviewer #3 (Comments to the Authors (Required)):

The Polycomb repressive complexes are important chromatin regulators found across eukaryotes. Although their function, composition and mechanisms of action have been intensively studied in many major model systems the evolution of the Polycomb system is not yet well understood. The authors here study the evolution of Polycomb Repressive Complex 2 (PRC2) components across eukaryotes. Their findings support the hypothesis that PRC2 emerged early in eukaryote evolution, likely being present in the last common eukaryotic ancestor and provide some additional support to the idea that E(z) may have first evolved as a dual K9/K27 methyltransferase. Although neither of these ideas are new and have been proposed by other studies this study is certainly valuable in providing additional evidence to support them. I think this manuscript is certainly worthy of publication but I have a number of issues and suggestions which I think should be addressed before acceptance:

1. My biggest concern with the manuscript is the quality of the data upon which the authors rely to make conclusions about presence/absence in different species and groups. The authors use BUSCO as a measure of the completeness of the genomic

information they use. It is not clear to me, however why having more than 50% completeness would be considered good. In fact, I would think by looking at the numbers for many of the genomes used that the BUSCO assessment shows that those genomes are very incomplete making it quite likely that potential homologs could be missed in those species. I think the authors should use some cut-off in whether to include a species in their analysis here at all. Some species have such a low score that they should not be included but unfortunately for some groups this would remove most of the species. This is particularly important when the authors want to make about about species or clades have no ortholog. Any decision on this will always be arbitrary but I would say that 50% is a reasonable cutoff.

This is not a major problem for the overall conclusions of the paper per se as even in groups where some genome are low quality the fact homologs are or are not found across the clade is good evidence. To put this another way, even if all the genome are low quality the fact it would be missed in many is low. It does however call into question some of the more specific claims made about presence/absence in specific species/clades. I think the authors should shy away from making major claims, particularly based on absence in species/clade where the quality of genomes is low, and should rather stick to more high level claims. For example, there are only 2 species in the Metamonada which have a BUSCO of higher than 50 and most have very low values which would lead to any claims from this group being less robust than for other groups. You see that within the Discoba the only two species which have an ESC or E(z) are the species with greater than 50. I would be interested if the authors were to recreate figure 1 now including only species with BUSCO > 50 or also with BUSCO > 75 what conclusions can now be drawn convincingly from the data.

2. I think the paper could be substantially shorter and many of the major points could be made in a more succinct way. At several points the authors describe in detail the presence/absence of homolog in various species which is totally unnecessary. This is particularly unimportant given the uncertainty over the quality of data for these species. It would make the paper much more readable if the authors were to make shorter more definitive statements of presence/absence in clades rather than including too much detail.

3. Throughout the manuscript the authors use the term "higher eukaryotes". This term is somewhat outdated and can be misleading. It is hard to define what higher means and it often suggests some relationship between groups that does not exist. I would try to replace such terms with more concise description of the clades/groups that you want to refer to.

4. Line 60: There is no need to use the term multicellular when referring to animals.

5. In several places the authors use the term "homolog" where it would be more appropriate to use the term "ortholog".

6. In a lot of places in the manuscript the writing is sloppy and it is unclear what the authors are talking about. For example, on line 90 a sentence ends "suggesting the existence.", and it seems the end of the sentence is missing. Similarly on line 92 the authors write "... a separate domain of life representing, ..." and again it seems like some text is missing as it is unclear what should come after representing. Some attention should be paid to the writing to make the paper more readable.

7. Although the authors focus on PRC2 a short description of what PRC1 and PRC2 are and what is known about PRC1 evolution would help orient the reader in the field. Several studies have addressed PRC1 evolution recently (Berke and Snel, 2015; Chen et al., 2016; Gahan et al., 2020; Schuettengruber et al., 2017) and a short description of what is known there may be useful. It also seems PRC2 is more conserved/older than PRC1 (or to be specific several components of PRC1) and this may be worth discussing.

8. In Figure 1 the authors use a very definite phylogeny of eukaryotes even though there has been considerable debate in this area. The authors should either state the source of the phylogeny they use and/or include some detail about possible alternative phylogenies.

9. Figure 1: In a couple of cases the colored icons have moved and are located outside of the circle.

10. The authors state that Suz(12) is absent in several groups. One of the species in which they claim it is missing is in Tetrahymena, however it has been shown that there is indeed a Suz12 in this species but it is substantially shorter than that found in other species (Xu et al., 2021). I assume that this was not counted as it did not meet the criteria set by the authors to assign orthology. Firstly, this should be discussed albeit briefly but secondly it also highlights an important point that in cases where loss is seen it may be that the protein has simply lost some domain or changed in some other way so as to be difficult to recognize as a true ortholog without additional experiments. This is a problem in all such studies and not something that the authors can help but a description of such possibilities would be helpful.

11. Line 137, you cannot start a new paragraph with "however" and in this case it makes no sense either way so please rephrase

12. Line 151-152 The authors state their ESC phylogeny is the "most robust ESC tree to date". This statement must somehow be backed up by data. It is not clear how many other ESC trees there are that the authors are talking about. These should be referenced. The authors must also state what they mean by "robust" and by which measurements they came to this conclusion. If this is simply the opinion of the authors, I would remove this statement entirely.

13. Line 152-154. The authors state "Similarly, unresolved relationships within microalgae species were reported (Zhao et al. 2020), whereas other studies suggest a monophyletic origin for ESC homologs found in most eukaryotes (Huang et al. 2017)." It is not clear to me however how the ESC phylogeny here fits with these reports. If the authors want to mention them here they should state if they find the same or different things to these two studies.

14. Line 157. The jump to NURF55 here is a bit abrupt and could do with starting a new paragraph.

15. Line 160. The reference here to possible horizontal gene transfer events is vague and not well supported. There are other explanations as to why a single gene family phylogeny may not recapitulate the phylogeny of the species used. The authors should be more reserved in any claims of horizontal gene transfer unless they have additional evidence for it. This is also the case later in the manuscript and the authors should be careful without to make claims about HGT without additional support from outside of the phylogeny.

16. Line 164-169. It is unclear to me what the authors are trying to say in this section. It looks as if some sentences are missing

which would make sense but as it is it doesn't fit together at all.

17. Line 212-243. This entire paragraph discusses in detail the clades into which E(z) homologs fall but in the end doesn't really reveal anything of importance to the paper and should be shortened down to a single statement.

18. Line 256. Just because sequences found in species close to the root does not necessarily suggest this has anything to do with the ancestral sequence and such statements should be avoided.

19. Line 299-301. The authors state: ". Our results show that the SET-domain proteins are less conserved in bacteria and archaea than what was reported previously (Alvarez-Venegas 2014), likely due to a lower number of representative species per group in our analysis." Given that the authors admit that the discrepancy in these studies is likely due to less species in their analysis that they cannot conclude that " SET-domain proteins are less conserved".

20. Line 393. The authors must cite the study which showed that paramecium E9z) is capable of methylating h3k9.

Berke, L., and Snel, B. (2015). The plant Polycomb repressive complex 1 (PRC1) existed in the ancestor of seed plants and has a complex duplication history. *BMC Evol Biol* 15, 44. 10.1186/s12862-015-0319-z.

Chen, D.H., Huang, Y., Ruan, Y., and Shen, W.H. (2016). The evolutionary landscape of PRC1 core components in green lineage. *Planta* 243, 825-846. 10.1007/s00425-015-2451-9.

Gahan, J.M., Rentzsch, F., and Schnitzler, C.E. (2020). The genetic basis for PRC1 complex diversity emerged early in animal evolution. *Proceedings of the National Academy of Sciences* 117, 22880-22889. 10.1073/pnas.2005136117.

Schuettengruber, B., Bourbon, H.M., Di Croce, L., and Cavalli, G. (2017). Genome Regulation by Polycomb and Trithorax: 70 Years and Counting. *Cell* 171, 34-57. 10.1016/j.cell.2017.08.002.

Xu, J., Zhao, X., Mao, F., Basrur, V., Ueberheide, B., Chait, B.T., Allis, C.D., Taverna, S.D., Gao, S., Wang, W., and Liu, Y. (2021). A Polycomb repressive complex is required for RNAi-mediated heterochromatin formation and dynamic distribution of nuclear bodies. *Nucleic Acids Res* 49, 5407-5425. 10.1093/nar/gkaa1262.

18.02.2022

Dear Dr. Novella Guidi, PhD

I am happy to submit a revised version of our manuscript “Phylogenetic profiling resolves early emergence of PRC-2 and illuminates its functional core” (#LSA-2021-01271-T) for consideration by Life Science Alliance. We have addressed all suggestions of the reviewers and provide point-by-point response to the comments raised by the reviewers. To ease the revision of the new version of the manuscript, all changes are mentioned in this rebuttal letter. In addition to the reviewers' suggestions, the manuscript format has also been changed to fit the LSA required format.

On behalf of all co-authors let me thank you for your and the reviewers' efforts invested into our manuscript, we are looking forward to hearing from you.

With best wishes

Abdoallah Sharaf

Reviewer #1:

The manuscript by Sharaf et al examines the evolutionary origin of the Polycomb Repressive Complex 2 (PRC2) using a phylogenetic approach. PRC2 deposits the histone mark H3K27me3 on silent protein-coding genes. The composition of the complex has been characterized in very few species and its enzymatic activity was experimentally tested in a handful of organisms. Here, the authors survey the genomes/transcriptomes of 283 species to search for 4 PRC2 core components: E(z), the catalytic subunit, ESC, SUZ12, and Nurf55. They identified E(z) and Esc homologs in species that support the emergence of PRC2 prior to the diversification of eukaryotes. Their analysis also suggests that the E(z) enzyme may have shifted substrate specificity- from K9 to K27 methylation- in the course of evolution.

Response: Thank you for your thorough review and comments that have helped to improve our manuscript. The major changes involve the addition of PcG-finder validation and additional analyses of VEFS-Box proteins across the studied species to consider higher diversification of Su(z)12 orthologs.

Major comments:

1. The authors developed a sequence homology-based computational automated tool (PcG-finder) for the search of PRC2 core components in genomes and transcriptomes, using *Drosophila melanogaster* PRC2 protein sequences. I wonder how robust and sensitive this tool is. Could the authors validate their tool by showing that they manage to recover all the known PRC2 core components in the organisms where PRC2 composition has been established experimentally: *Arabidopsis*, *C. elegans*, *Neurospora crassa*, *Cryptococcus neoformans*, *Paramecium tetraurelia*, *Tetrahymena thermophila*.

2. Along the same line, SUZ12 is not always found in PRC2 complexes. In the fungus *Cryptococcus neoformans* (Dumesic, 2015) and in *C. elegans* (Bender 2004; Xu 2001), no obvious SUZ12 homologs were identified, yet *Cryptococcus* Bnd and *C. elegans* MES3, which are required for the catalytic activity of the complex, might be functional homologs of SUZ12, as suggested before for MES3 (Schuettengruber, 2017). Do the authors identify those proteins with their tool?

Pull down experiments recently revealed the existence of putative SUZ12 components in PRC2 in the ciliates *Paramecium tetraurelia* (Miro Pina 2021 <https://doi.org/10.1101/2021.08.12.456067>) and *Tetrahymena thermophila* (Xu 2021 <https://doi.org/10.1093/nar/gkaa1262>) that could not be identified by homology-based search because of sequence divergence.

The conclusion that there is evidence of the absence of SUZ12 in ciliates (Figure 1- Figure 5- line 143) should thus be corrected. In addition, the limitations of the approach presented in the manuscript should be discussed. It might also be wise not to draw any conclusions on the dispensability of SUZ12 based on its absence. Instead, I suggest considering the alternative possibility that SUZ12 might evolve more rapidly than the other components.

The authors should rephrase the abstract (Lines 21-22) and the results section (lines 143- lines 279-281).

Response to comments 1 and 2:

We have performed the validation of the computational pipeline in the first steps of our work but have not included the results in the original version of the manuscript, which was a clear mistake on our side. In the revised version, we newly include a first chapter of results entitled “Specificity and sensitivity of PcG-finder, an automated protein homolog identification pipeline” and Table S1 where we summarize the results of the pipeline validation. The PcG-finder tool has been validated comparing its predictions of all the PRC2 core components in 8 organisms from different eukaryotic lineages, including the species mentioned by the reviewer. We find that the pipeline can selectively identify sequence homologs in all the organisms except for the Su(z)12 proteins in *N. crassa* and *T. thermophila* and ESC in ciliates, that carry limited sequence conservation. Importantly, all the known E(z) orthologs are faithfully recovered by the pipeline. Being a sequence-homology-based pipeline, PcG-finder will not detect PRC2 core subunits that are functional but not sequence homologs and that do not carry characteristic protein domains – i.e. Su(z)12 functional homologs in *C. elegans*, *C. neoformans* or *P. tetraurelia*. In the revised version, we are discussing the limitations of the approach to avoid misinterpretation of homolog absence.

We would like to thank the reviewer for highlighting the divergence instead of loss of Su(z)12 subunit. Based on this comment and not having identified the VEFS-Box-containing Su(z)12 orthologs in *N. crassa* and *T. thermophila*, we included a domain-based protein search to identify all VEFS-Box proteins in the species studied and the results have been included in the revised version of the manuscript. Using this approach, we identify the Su(z)12 orthologs in the above-mentioned species, indeed justifying the approach to search for more divergent Su(z)12-like proteins. We are now basing our conclusions and discussion on the combined results of the two approaches. Therefore, all conclusions on the dispensability of Su(z)12 based on its absence have been removed or rephrased to accommodate the possibility of more divergent VEFS-Box proteins or even unrelated proteins fulfilling the function of Su(z)12. We think that expanding the manuscript by providing the data on VEFS-Box proteins (Fig 1 and fully listed in Table S8) is a useful extension for the scientific community, although proteins not involved in PRC2 (false positives) are likely to be identified this way. In this version of the manuscript, we have distinguished Su(z)12 orthologs with sufficient protein sequence similarity to be identified by PcG-finder, Su(z)12 orthologs that can only be identified based on the existence of the VEFS domain (e.g. *T. thermophila*, *N. crassa*) and proteins without sequence or functional domain conservation that may be functional homologs of Su(z)12 – e.g. Bnd1 in *C. neoformans* or MES2 in *C. elegans*, which cannot be identified by homology search. We call these “functional homologs” of Su(z)12 and do not include them among Su(z)12 or VEFS-Box proteins.

Based on the additional results, we have rephrased the Su(z)12-relevant sections as follows:

Abstract:

FROM: *"Analyzing 283 species, we robustly detect a common presence of E(z) and ESC, suggesting that Su(z)12 may have emerged later and/or maybe dispensable from the evolutionarily conserved functional core of PRC2."*

TO: *"Full-length Su(z)12 orthologs were identified in some lineages and species only, indicating, non-exclusively, high divergence of VEFS-Box-containing Su(z)12-like proteins, functional convergence of sequentially unrelated proteins or Su(z)12 dispensability."*

Original line 143:

FROM: *"Our results suggest the Su(z)12 was lost in Alveolata at the root of Myzozoa."*

TO: Lines 190-192: *"Su(z)12 is generally absent in Alveolata except for the recently described Colponemidia clade (Tikhonenkov et al. 2020), suggesting divergence of the Su(z)12 at the root of Myzozoa."*

Original lines 279-281:

FROM: *"Su(z)12 may have emerged later and its secondary loss seems more frequent than loss of ESC, suggesting its dispensability."*

TO: Lines 328-342: *"We have not found conserved Su(z)12 orthologs in multiple lineages including Discoba and Metamonada (Fig 1 and Supplementary file 1). VEFS-Box proteins are nevertheless present in most of these lineages (Fig 1 and Table S8) and they may represent more divergent Su(z)12-like proteins. VEFS-Box proteins are found in alveolates where E(z) homologs seem missing (Fig 1). We found non-Su(z)12 VEFS-Box proteins in model species with well-described Su(z)12 orthologs, such as human, D. melanogaster or A. thaliana, indicating potential engagement of VEFS-Box proteins in other complexes. Experimental evidence will be needed to associate VEFS-Box proteins with PRC2. It is interesting to note that the currently described examples of PRC2 where other than VEFS-box proteins take the function of Su(z)12 come from species (C. elegans, C. neoformans and P. tetraurelia) where no VEFS-Box proteins were identified (Table S8) and that contain ancestral E(z) of Clades I or II. Similarly, highly divergent Su(z)12 (or even other VEFS-Box proteins) are involved in PRC2 in fungi and ciliate T. thermophila within Clade I and/or Clade II E(z). This may indicate later evolution and engagement of VEFS-box proteins and canonical Su(z)12 in PRC2, but this hypothesis must be addressed experimentally."*

Minor comments:

- Line 13: "maintaining" not "establishing transcriptionally silent chromatin states"

Response: Thank you, it is a valid criticism and the word "establishing" has been replaced by "maintaining".

- Lines 50-53: regarding the evidence for H3K27me3 in unicellular eukaryotes, some references are missing (eg Carlier, 2021; Jamieson 2013, Lhuillier-Akakpo 2014; Liu 2007)

Response: Thank you and excuse us for the omission. The references Lhuillier-Akakpo 2014 and Liu 2007 have been added. The references to Carlier 2021 (*Podospira anserina*) and Jamieson

2013 (*Neurospora crassa*) have not been added as the sentence relates to unicellular species rather than filamentous (i.e. simple multicellular) species.

- Lines 68-69: The association between H3K27me3 and genomic repeats could be supported by a more complete list of references and/or a review.

Response: Thank you for the comment, we have lost some crucial references while shortening the text. We have newly added the references to reviews that better reflect and provide discussion related to H3K27me3 in constitutive heterochromatin (Deleris et al.2021, Vijayanathan et al 2022).

- Lines 90-91: The sentence is incomplete: "suggesting the evidence..."

Response: Thank you for noticing and please excuse the negligence. The sentence has been completed.

- Lines 132-134: The statement regarding NURF55 function outside PRC2 is correct. It would be better to state that NURF55 belongs to several complexes earlier in the introduction.

Response: We have included the following statement in the introduction:

Line 73-74: "*NURF55 is involved in multiple chromatin-related protein complexes (Hennig et al. 2005; Sukanuma et al. 2008).*"

- Line 220: reference to E(z) phylogeny in ciliates should be corrected. It is not Frapporti 2019 but Lhuillier-Akakpo 2014.

Response: Thank you for the correction, the reference has been changed. The whole passage is newly included in Supplementary file 1.

- Lines 329-332. The sentence is unclear. Why is this a good model for studying the evolution of PRC2 catalytic function?

Response: We have removed the statement. Current version: Lines 369-371: "*The E(z) subfamily of SET-domain proteins therefore seems to have diverged after the emergence of eukaryotes but prior to their expansion.*"

- Lines 387-389: Regarding the structural modeling of SET domains of E(z) proteins belonging to the 5 clades, the authors state that clade II is different from the others: "some of the loops/turns near the cofactor binding pocket are slightly disordered when compared to other models, which might be the reason for the binding difference". Could the authors be more precise: I don't understand from their description what exactly differs in clade II enzyme and how this is linked to substrate specificity? Clade II Paramecium tetraurelia SET domain has been modeled in a previous study (Frapporti 2019). How similar/different are the two modeled structures and conclusions?

Response: We have rephrased the paragraph.

Lines 419-422: “The representative protein model for clade II (*P. tetraurelia*) is comparable to the previously computed model (Frapporti et al., 2019), although the amino acids comprising the C-terminal post-SET domain are not included in our model. Importantly, structural folds are conserved, and retain similar structural topology and substrate binding patterns (Fig S8).”

In our model (Fig 4), the N-terminal alpha-helix appears shorter than in the model of Frapporti et al 2019. In our study, we used the SET-domain of EZH1 (PDB ID: 7KSR, EZH1—identity- 38.66%) as a reference structure, while Frapporti et al. use SET-domain of EZH2. To respond to this comment, we modelled the *P. tetraurelia* SET-domain using EZH2 (PDB ID: 5hyn.1.A--EZH2--36.97%) as a reference template. In this case, the helix is longer and the model resembles the published one. We do not see the post-SET domain structure as this sequence was not included in our modelled sequence. Importantly, the substrate orientation is similar in both models, showing K27 oriented in proximity of the cofactor. (Superimposed models obtained by this study are depicted on the left: based on EZH1 in pink, based on EZH2 in green, Frapporti et al.2019 model on the right)

- Lines 392-394: add the reference.

Response: The reference to Frapporti et al. 2019 has been added – we apologize for missing it.

Lines 425-427: “Previously, H3K9me (as well as H3K27me) was experimentally shown to be carried out by the Ezi homolog of *P. tetraurelia* (clade II) (Frapporti et al. 2019).”

- Lines 415/416: what do the authors mean by "biological significance of PRC2": its biological functions?

Response: We have revised the sentence. Although genomic and biological functions clearly need to be addressed, this is not specifically highlighted by the presented work. We have therefore rephrased to: Lines 447-450: *“However, our findings highlight important future questions that need to be addressed experimentally, including PRC2 catalytic activity, E(z) substrate specificity, or involvement of VEFS-Box proteins and other functional homologs of Su(z)12.”*

- Figure 5: A "parasitism lifestyle" cannot be used to describe Ciliophora/Alveolata, as indicated in the legend. This should be corrected.

Response: Thank you, the lifestyle of ciliophora has been corrected.

Reviewer #2:

Polycomb Repressive Complex 2 (PRC2) has been shown to play crucial roles in cell differentiation and cell fate maintenance during the development of many metazoan models by generating specific histone modifications and establishing a poised gene expression program. In the article, "Phylogenetic profiling resolves early emergence of PRC2 and illuminates its functional core", the authors performed comprehensive phylogenetic analysis to study the distribution of PRC2 components across eukaryotes. Upon surveying the sequence information of 283 species, the authors identified a putative ortholog of Ez in a subset of Discoba, an earlier eukaryotic lineage. The authors did not observe Ez in Metamonada, although both Discoba and Metamonada belong to the Excavates superfamily. After studying distribution, the authors focused on the Ez component, which harbors the enzymatic SET domain thus compose a catalytic part of the PRC2. The authors classified full-length Ez into five groups and identified a group where Discoba is located with other species (group II). Then, the authors zoomed into SET-domain and further compared domain structures across five groups. In a structure model, the histone substrate-binding pocket of group II showed more disordered features than other groups.

Overall, this study represents comprehensive phylogenetic profiling of the PRC2 subunits across the entire Eukaryota. My main concern is the presentation, in particular Figures 1-3, needs to be simplified and up to the point.

Response: Thank you for your thoughtful and thorough review. Major changes involve modification of figures and rearrangement of main text and supplementary figures. We have simplified the font (sans-serif) for all figures and only kept the species names (avoiding accession numbers) to increase legibility of figures 2 and 3. In case of both Figures 2 and 3, we created new version of the figure, collapsing leaves including species from the major supergroups where possible. This way, the association of the clades with the major supergroups is more obvious. Based on comments of Reviewer 1 and Reviewer 3, we have added a section on PcG finder validation and analyses of the distribution of VEFS-Box proteins in the studied species.

1. Figure 1: Manuscript (written part of Figure 1) is relatively straightforward, but Figure 1 itself is too complicated to understand by looking at it. Figure legend is poorly described and contains little information. What are the main messages of the figure? Where is that info? There are many unnecessary parts of the figure as well. For example, what is the added value of the numbers next to the species? Species names are barely readable on a 100% scale. Do you need all those names and numbers in the main figure? The authors should move the current version into supplementary and provide a simple figure. Also, it would be helpful to add a photo or image of each representative species used in their analysis.

Response: Thank you for these suggestions. We fully agree with the reviewer that the original image was complex. We have tried to simplify the image: we have simplified the font (sans-serif) and only kept the species names (avoiding accession numbers) to increase legibility. Only species with BUSCO score 50 are now plotted (where BUSCO score 75 and higher is indicated in bold) and complete data is available in the supplement. We believe that the new version of the figure is more legible. At the same time, we would prefer to maintain the information contained in the current version of the image – although we have considered options of collapsing the groups, we feel that this may result in misleading information due to the variability within the groups. The summary of the findings is provided in Figure 5 (including representative species images) and we would prefer to keep this revised version of Fig 1 as an overview of the study, if possible.

2. Among the PRC2 components, NURF55 shows the most substantial conservation across eukaryotes. Nevertheless. The authors focused on Ez. The rationale for Ez selection has to be apparent from the beginning of the manuscript.

Response: Thank you, the introduction has been modified in the following ways:

Lines 73-76: “*NURF55 is involved in multiple chromatin-related protein complexes (Hennig et al. 2005; Saganuma et al. 2008). Thus, only E(z), ESC, and Su(z)12 subunit were used to infer the phylogeny of the PRC2 complex to bypass confusing interpretations (Huang et al. 2017; Shaver et al. 2010).*” and Lines 86-87: “*The catalytic activity of PRC2 is carried out by E(z), which is therefore the defining functional subunit.*”

3. Figures 2 and 3 have the same issue as Figure 1. The numbers in triangles and branches are too small to recognize and may not be necessary for the main figure. Figure 2A - Information about different species in each clade is an interesting biological outcome, but it is hard to get from the figure. Figure 2B - it is unclear what is the take-home message of this figure. The written part (page 7; 251-272) is also confusing.

Response: Thank you. We have simplified the font (sans-serif) for all figures and only kept the species names (omitting accession numbers) to increase legibility in Figures 2&3. In case of both Figures 2 and 3, we created new versions of the figure, collapsing leaves including species from the major supergroups where possible. This way, the association of the clades with the major supergroups is more obvious. We have in addition removed panel A depicting the domain organization and keep this as supplementary Fig S5.

Finally, the written part (original page 7; 251-272) has been simplified to:

Lines 314-319: *“Clade I and II proteins contain either only the SET domain or sequentially arranged CXC and SET domains (Fig S5 and Table S10). The SANT domain is present in clades III-IV N-terminally of the arranged canonical domains (CXC and SET) (Fig S5 and Table S10). More complex domain architectures were observed within proteins of clade V. This clade contains all plant and animal sequences, including eumetazoan orthologs (EZH1 and EZH2) (Fig S5 and Table S10).”*

4. I understand the challenge of assembling and organizing all info after sequencing analysis. However, the current version of the figure set seems premature. These figures show the crude output of the research (species names, conservation scores, etc.), and the authors made a little effort to deliver biological meaning to more broad and general readers.

Response: Thank you, it was a valid and valuable comment that prompted us to modify the figures to hopefully better convey the main message. We tried our best to simplify but not to lose information that we believe is important.

5. Figure 4 is excellently delivered the biological message. Figure 5 is constructive to get the summary view of Figure 1-3. It would be helpful to add info about the percentage value of the existing PRC2 components. For instance, most Aveloata they observed do not contain Suz12, ESC, and Ez.

Response: To this end, we have added values representing the percentage of species where PRC2 components were found next to the symbols for the respective subunits. In the legend, we further refer to the relevant supplementary table S2 where the full list and numbers of species studied can be found.

Reviewer #3

The Polycomb repressive complexes are important chromatin regulators found across eukaryotes. Although their function, composition and mechanisms of action have been intensively studied in many major model systems the evolution of the Polycomb system is not yet well understood. The authors here study the evolution of Polycomb Repressive Complex 2 (PRC2) components across eukaryotes. Their findings support the hypothesis that PRC2 emerged early in eukaryote evolution, likely being present in the last common eukaryotic ancestor and provide some additional support to the idea that E(z) may have first evolved as a dual K9/K27 methyltransferase. Although neither of these ideas are new and have been proposed by other studies this study is certainly valuable in providing additional evidence to support them. I think

this manuscript is certainly worthy of publication but I have a number of issues and suggestions which I think should be addressed before acceptance:

Response: Thank you for your insightful comments and thorough review. We have tried to address each of the comments as detailed below. The major changes involve modification of figures and rearranging the information included in the main text vs. supplement. Based on the Reviewer 1 and Reviewer 3 comments, we have added a section on PcG finder validation and analyses of the distribution of VEFS-Box proteins in the studied species.

1. My biggest concern with the manuscript is the quality of the data upon which the authors rely to make conclusions about presence/absence in different species and groups. The authors use BUSCO as a measure of the completeness of the genomic information they use. It is not clear to me, however why having more than 50% completeness would be considered good. In fact, I would think by looking at the numbers for many of the genomes used that the BUSCO assessment shows that those genomes are very incomplete making it quite likely that potential homologs could be missed in those species. I think the authors should use some cut-off in whether to include a species in their analysis here at all. Some species have such a low score that they should not be included but unfortunately for some groups this would remove most of the species. This is particularly important when the authors want to make about about species or clades have no ortholog. Any decision on this will always be arbitrary but I would say that 50% is a reasonable cutoff.

This is not a major problem for the overall conclusions of the paper per se as even in groups where some genome are low quality the fact homologs are or are not found across the clade is good evidence. To put this another way, even if all the genome are low quality the fact it would be missed in many is low. It does however call into question some of the more specific claims made about presence/absence in specific species/clades. I think the authors should shy away from making major claims, particularly based on absence in species/clade where the quality of genomes is low, and should rather stick to more high level claims. For example, there are only 2 species in the Metamonada which have a BUSCO of higher than 50 and most have very low values which would lead to any claims from this group being less robust than for other groups. You see that within the Discoba the only two species which have an ESC or E(z) are the species with greater than 50. I would be interested if the authors were to recreate figure 1 now including only species with BUSCO > 50 or also with BUSCO>75 what conclusions can now be drawn convincingly from the data.

Response: Thank you for this helpful suggestion. We have simplified Figure 1, plotting only species with BUSCO score 50 and in bold indicating those with BUSCO score 75 and higher.

We have the following arguments for keeping species with BUSCO score 50:

- Stramenopile and Alveolate species mainly have BUSCO score 50, still the contrasting success of identifying PRC2 components in the two supergroups is evident. The same

holds true for contrasting success of identification of PRC2 components in Stramenopile groups – in particular comparing diatoms with ochrophytes/chrystophytes.

- Sometimes, PRC2 homologs are found in species with BUSCO50 but not in species with BUSCO75, which could create a misleading picture of the group – e.g. Glaucocystophyceae.
- One of the main findings is the identification of E(z) in Discoba – e.g. Euglena. Including two Euglena species where E(z) homologs are identified despite different BUSCO scores (E. longa – BUSCO75 and E. gracilis – BUSCO 50) strengthens our conclusion in our opinion.

2. I think the paper could be substantially shorter and many of the major points could be made in a more succinct way. At several points the authors describe in detail the presence/absence of homolog in various species which is totally unnecessary. This is particularly unimportant given the uncertainty over the quality of data for these species. It would make the paper much more readable if the authors were to make shorter more definitive statements of presence/absence in clades rather than including too much detail.

Response: Thank you, statements especially related to absence of the subunits were removed to avoid misinterpretation stemming from uncertain data quality. Detailed information was either removed (if unnecessary) or moved to the supplementary file. We believe this made the manuscript more concise and legible.

3. Throughout the manuscript the authors use the term "higher eukaryotes". This term is somewhat outdated and can be misleading. It is hard to define what higher means and it often suggests some relationship between groups that does not exist. I would try to replace such terms with more concise description of the clades/groups that you want to refer to.

Response: Thank you, yes we realize the problem of using the term "higher or early eukaryotes" and it has now been replaced by the clade or group names throughout the manuscript.

4. Line 60: There is no need to use the term multicellular when referring to animals.

Response: We have removed the term "multicellular".

5. In several places the authors use the term "homolog" where it would be more appropriate to use the term "ortholog".

Response: Thank you, we have revised the manuscript for the use of the terms "homolog" and "ortholog".

6. In a lot of places in the manuscript the writing is sloppy and it is unclear what the authors are talking about. For example, on line 90 a sentence ends "suggesting the existence.", and it seems the end of the sentence is missing. Similarly on line 92 the authors write "... a separate domain of life representing, ..." and again it seems like some text is missing as it is unclear what should

come after representing. Some attention should be paid to the writing to make the paper more readable.

Response: We apologise for these careless mistakes that were caused in the final stages of the manuscript revisions. The sentences have been completed and the manuscript has been carefully revised.

7. Although the authors focus on PRC2 a short description of what PRC1 and PRC2 are and what is known about PRC1 evolution would help orient the reader in the field. Several studies have addressed PRC1 evolution recently (Berke and Snel, 2015; Chen et al., 2016; Gahan et al., 2020; Schuettengruber et al., 2017) and a short description of what is known there may be useful. It also seems PRC2 is more conserved/older than PRC1 (or to be specific several components of PRC1) and this may be worth discussing.

Response: Here, we have added the following sentences in the introduction to highlight the difference in conservation of PRC1 and PRC2:

Lines 45-53: *“The best-well studied PcG complexes are represented by the ubiquitin-ligase Polycomb Repressive Complex 1 (PRC1) and the histone methyltransferase PRC2, that establish histone 2A ubiquitination (H2Aub) and H3 lysine 27 methylation (H3K27me), respectively. PRC1 is hypothesized to have emerged through convergent evolution, as its core subunits differ in animals (Gahan et al. 2020) and in plants (Berke and Snel 2014; Chen et al. 2016), despite conserved catalytic activity (Calonje 2014; Schuettengruber et al. 2017). In contrast, PRC2 is hypothesized to have diverged early in eukaryotic evolution and its core subunits are generally conserved in animals, plants and other major eukaryotic lineages (Shaver et al. 2010; Baile et al. 2021).”*

8. In Figure 1 the authors use a very definite phylogeny of eukaryotes even though there has been considerable debate in this area. The authors should either state the source of the phylogeny they use and/or include some detail about possible alternative phylogenies.

Response: In the revised version of Figure 1 (and Figure 5) legend, we have cited the source of the phylogeny.

9. Figure 1: In a couple of cases the colored icons have moved and are located outside of the circle.

Response: Thank you for noticing, Figure 1 has been modified and revised, together with the graphics.

10. The authors state that Suz(12) is absent in several groups. One of the species in which they claim it is missing is in Tetrahymena, however it has been shown that there is indeed a Suz12 in this species but it is substantially shorted that found in other species (Xu et al., 2021). I assume that this was not counted as it did not meet the criteria set by the authors to assign orthology. Firstly, this should be discussed albeit briefly but secondly it also highlights an important point that in cases where loss is seen it may be that the protein has simply lost some domain or changed

in some other way so as to be difficult to recognize as a true ortholog without additional experiments. This is a problem in all such studies and not something that the authors can help but a description of such possibilities would be helpful.

Response: Thank you for this relevant comment that has also been raised by Reviewer 1 as a major concern. Based on these comments, we addressed the sequence divergence of Su(z)12 orthologs in ciliates and have analyzed the distribution of VEFS-Box proteins to indicate potential candidate Su(z)12-like homologs. We think this could also be a useful resource for the research community to aid future experimental validation of the candidate PRC2 subunits. We have extended the discussion in the section "Phylogenetic distribution of putative PRC2 subunit orthologs" to avoid making potentially misleading conclusions related to the absence of PRC2 components. All conclusions on the dispensability of Su(z)12 based on its absence have been removed or rephrased.

11. Line 137, you cannot start a new paragraph with "however" and in this case it makes no sense either way so please re-phrase

Response: Thank you, the sentence has been rephrased.

12. Line 151-152 The authors state their ESC phylogeny is the "most robust ESC tree to date". This statement must somehow be backed up by data. It is not clear how many other ESC trees there are that the authors are talking about. These should be referenced. The authors must also state what they mean by "robust" and by which measurements they came to this conclusion. If this is simply the opinion of the authors, I would remove this statement entirely.

Response: Indeed, this is a valid comment. We have rephrased the sentence to remove the statement.

13. Line 152-154. The authors state "Similarly, unresolved relationships within microalgae species were reported (Zhao et al. 2020), whereas other studies suggest a monophyletic origin for ESC homologs found in most eukaryotes (Huang et al. 2017)." IT is not clear to me however how the ESC phylogeny here fits with these reports. If the authors want to mention them here they should state if they find the same or different things to these two studies.

Response: In the new version of the manuscript, we have moved this part of the text into the Supplementary file 1 and the sentence has been rephrased.

14. Line 157. The jump to NURF55 here is a bit abrupt and could do with starting as a new paragraph.

Response: In the revised version, NURF55 results are now discussed in a new paragraph (lines 245-255).

15. Line 160. The refence here to possible horizontal gene transfer events is vague and not well supported. There are other explanations as to why a single gene family phylogeny may not recapitulate the phylogeny of the species used. The authors should be more reserved in any

claims of horizontal gene transfer unless they have additional evidence for it. This is also the case later in the manuscript and the authors should be careful without to make claims about HGT without additional support from outside of the phylogeny.

Response: In the revised version of the manuscript, the statements have been removed.

16. Line 164-169. It is unclear to me what the authors are trying to say in this section. It looks as if some sentences are missing which would make sense but as it is it doesn't fit together at all.

Response: We have shortened the paragraph describing NURF55 results (now lines 245-255) and moved extended explanation into the Supplementary file 1. We hope that the text is more understandable this way.

17. Line 212-243. This entire paragraph discusses in detail the clades into which E(z) homologs fall but in the end doesn't really reveal anything of importance to the paper and should be shortened down to a single statement.

Response: Thank you, the paragraph has been rephrased and unnecessary details moved to Supplementary file 1.

18. Line 256. Just because sequences found in species close to the root does not necessarily suggest this has anything to do with the ancestral sequence and such statements should be avoided.

Response: We have removed the statements from the revised main text as well as from the Supplementary file 1 that contains extended result description.

19. Line 299-301. The authors state: ". Our results show that the SET-domain proteins are less conserved in bacteria and archaea than what was reported previously (Alvarez-Venegas 2014), likely due to a lower number of representative species per group in our analysis." Given that the authors admit that the discrepancy in these studies is likely due to less species in their analysis that they cannot conclude that " SET-domain proteins are less conserved".

Response: Yes, this is a valid comment and we have removed the statement.

20. Line 393. The authors must cite the study which showed that paramecium E9z) is capable of methylating h3k9.

Response: We apologise for the omission, the reference has been added: Lines 425-427: "*Previously, H3K9me (as well as H3K27me) was experimentally shown to be carried out by the Ezi homolog of P. tetraurelia (clade II) (Frapporti et al. 2019)*"

March 15, 2022

RE: Life Science Alliance Manuscript #LSA-2021-01271-TR

Dr. Abdoallah Sharaf
Biology Centre
Plant Epigenetics
Branišovská 31
České Budějovice, NA 37005
Czech Republic

Dear Dr. Sharaf,

Thank you for submitting your revised manuscript entitled "Phylogenetic profiling resolves early emergence of PRC 2 and illuminates its functional core". We would be happy to publish your paper in Life Science Alliance pending final revisions necessary to meet our formatting guidelines.

- Please address the remaining Reviewer 1 and 3 points
- please upload all figure files as individual ones, including the supplementary figure files
- please add the Twitter handle of your host institute/organization as well as your own or/and one of the authors in our system
- please separate the Results and Discussion section into two - 1. Results 2. Discussion, as per our formatting requirements
- please add callouts for Figure S9A-D to your main manuscript text

A. FINAL FILES:

B. MANUSCRIPT ORGANIZATION AND FORMATTING:

Sincerely,

Reviewer #1 (Comments to the Authors (Required)):

The authors have addressed all my concerns. The revised manuscript is much easier to read. I support the publication of this work in Life Science Alliance.

I just have a few minor comments:

- Figure 1: As said before, Esc homologs have been identified in T thermophila and P tetraurelia and are not represented in the figure. The title of Figure 1 should specify that the represented data come from PcG-finder.
- lines 152-155. Please rephrase the sentence, which is not clear to me. Does "sequentially" here mean sequence unrelated (divergent) proteins?
- Line 333 typo "described"
- Lines 340-341: please rephrase "This may indicate later evolution and engagement of VEFS-box proteins and canonical Su(z)12 in PRC2". I do not understand the meaning.

Reviewer #3 (Comments to the Authors (Required)):

The authors have addressed my comments sufficiently and the manuscript is much improved. There are just a few spelling and grammatical errors which I expect would be dealt with during copy editing and therefore I recommend that it be published.

21.03.2022

Dear Dr. Novella Guidi, PhD

I am happy to submit a revised version of our manuscript “Phylogenetic profiling resolves early emergence of PRC2 and illuminates its functional core” (#LSA-2021-01271-TRR) for consideration by Life Science Alliance. We have addressed all suggestions of the reviewers and provide point-by-point response to the comments raised by the reviewers. To ease the revision of the new version of the manuscript, all changes are mentioned in this cover letter. In addition to the reviewers' suggestions.

On behalf of all co-authors let me thank you for your and the reviewers' efforts invested into our manuscript, we are looking forward to hearing from you.

Reviewer #1:

- Figure 1: As said before, Esc homologs have been identified in *T thermophila* and *P tetraurelia* and are not represented in the figure. The title of Figure 1 should specify that the represented data come from PcG-finder.

Response: Thank you for these suggestions, The title of Figure 1 has been edited.

“Figure 1 - Species cladogram showing the distribution of PRC2 subunit orthologs identified using PcG-finder across eukaryotic lineages.”

- lines 152-155. Please rephrase the sentence, which is not clear to me. Does "sequentially" here mean sequence unrelated (divergent) proteins?

Response: Thank you, We have rephrased the sentence.

"In cases where divergent (sequence unrelated) proteins fulfill the function of canonical PRC2 subunits, such as is the case for suggested Su(z)12 functional orthologs (Dumesic et al. 2015; Bender et al. 2004), these proteins are likely to be absent in our search and need to be identified experimentally. "

- Line 333 typo "described"

Response: Thank you, it is a valid criticism, and the typo has been corrected.

- Lines 340-341: please rephrase "This may indicate later evolution and engagement of VEFS-box proteins and canonical Su(z)12 in PRC2". I do not understand the meaning.

Response: We have rephrased the sentence.

"This may indicate that other than VEFS-box proteins may be engaged in PRC2 complexes that contain ancestral E(z), but this hypothesis must be addressed experimentally."

Reviewer #3:

The authors have addressed my comments sufficiently and the manuscript is much improved. There are just a few spelling and grammatical errors which I wxpect would be dealt with during copy editing and therefore I reccommnd that it be published.

Response: Thank you for noticing and please excuse the errors. The manuscript has been carefully revised.

March 23, 2022

RE: Life Science Alliance Manuscript #LSA-2021-01271-TRR

Dr. Abdoallah Sharaf
Biology Centre
Plant Epigenetics
Branišovská 31
České Budějovice, NA 37005
Czech Republic

Dear Dr. Sharaf,

Thank you for submitting your Research Article entitled "Phylogenetic profiling resolves early emergence of PRC2 and illuminates its functional core". It is a pleasure to let you know that your manuscript is now accepted for publication in Life Science Alliance. Congratulations on this interesting work.

DISTRIBUTION OF MATERIALS:

Again, congratulations on a very nice paper. I hope you found the review process to be constructive and are pleased with how the manuscript was handled editorially. We look forward to future exciting submissions from your lab.

Sincerely,
